# Impacts of human mobility on the citywide transmission dynamics of 18 respiratory viruses in pre- and post-COVID-19 pandemic years

Amanda C. Perofsky [1,2] ✉, Chelsea L. Hansen [1,2,3], Roy Burstein [4], Shanda Boyle [1], Robin Prentice [1], Cooper Marshall [1], David Reinhart[1], Ben Capodanno [1], Melissa Truong[1], Kristen Schwabe-Fry[1], Kayla Kuchta[1], Brian Pfau[1], Zack Acker[1], Jover Lee[5], Thomas R. Sibley [5], Evan McDermot [1], Leslie Rodriguez-Salas[1], Jeremy Stone[1], Luis Gamboa[1], Peter D. Han[1,6], Amanda Adler[7], Alpana Waghmare[5,7,8], Michael L. Jackson[9], Michael Famulare[4], Jay Shendure [1,6,10], Trevor Bedford[1,5,6,10], Helen Y. Chu [11], Janet A. Englund [1,7,8], Lea M. Starita [1,6] & Cécile Viboud [2]

Many studies have used mobile device location data to model SARS-CoV-2 dynamics, yet relationships between mobility behavior and endemic respiratory pathogens are less understood. We studied the effects of population mobility on the transmission of 17 endemic viruses and SARS-CoV-2 in Seattle over a 4-year period, 2018-2022. Before 2020, visits to schools and daycares, within-city mixing, and visitor inflow preceded or coincided with seasonal outbreaks of endemic viruses. Pathogen circulation dropped substantially after the initiation of COVID-19 stay-at-home orders in March 2020. During this period, mobility was a positive, leading indicator of transmission of all endemic viruses and lagging and negatively correlated with SARS-CoV-2 activity. Mobility was briefly predictive of SARS-CoV-2 transmission when restrictions relaxed but associations weakened in subsequent waves. The rebound of endemic viruses was heterogeneously timed but exhibited stronger, longer-lasting relationships with mobility than SARS-CoV-2. Overall, mobility is most predictive of respiratory virus transmission during periods of dramatic behavioral change and at the beginning of epidemic waves.

In early 2020, there was widespread adoption of public health measures to slow the spread of severe acute respiratory syndrome coronavirus 2 (SARS-CoV-2). A variety of non-pharmaceutical interventions (NPIs) were implemented in most countries to reduce contact between infected and susceptible individuals, including shelter-in-place or stay-at-home orders, gathering restrictions, school and business closures, and travel bans. These interventions were effective at reducing not only SARS-CoV-2 transmission but also the spread of other directly transmitted respiratory pathogens[1–9]. Many endemic respiratory viruses did not return to widespread circulation until the end of 2020 or 2021[3,8–10], coinciding with the gradual lifting of social distancing measures and mask mandates.

During the COVID-19 pandemic, aggregated location data from mobile phones became an important source of information on

changes in population-level movements and the effectiveness of NPIs on SARS-CoV-2 transmission[11]. However, few studies have explored relationships between human mobility and the dynamics of endemic respiratory pathogens during the pandemic. Here we define "endemic" pathogens as those that have regular periodic cycles and stable rates of infection in outbreak periods. Due to the lack of circulation of endemic respiratory viruses in the first years of the pandemic, population susceptibility to these pathogens is expected to have increased, leading to earlier, larger, or more severe epidemics a few months later[12–14]. Understanding the influence of mobility patterns on the dynamics of endemic pathogens is important for predictive purposes, especially as perturbed circulation can lead to overlapping epidemics of different pathogens and, in turn, put extreme pressure on the healthcare system (e.g., the US "tripledemic" during winter 2022–2023)[15].

Here, we investigate the impacts of population behavior on the transmission dynamics of respiratory viruses in the greater Seattle, Washington region over a 4-year period, November 2018 to June 2022, using uniquely detailed data from hospital- and community-based respiratory surveillance and the collective movements of mobile device users. To determine which behavioral indices may best capture transmission-relevant contacts, we systematically relate changes in the daily transmissibility of 18 common respiratory viruses to trends in within-city mixing, visitor inflow, the percentage of devices staying home, foot traffic to various categories of points of interest (POIs), masking, and COVID-19 NPIs. These viruses span different transmission modes, seasonal cycles, and age distributions of infection and include SARS-CoV-2, influenza viruses (A/H3N2, A/H1N1, and B), respiratory syncytial viruses (RSV A and B), seasonal coronaviruses (hCoV 229E, OC43, HKU1, and NL63), human metapneumovirus (hMPV), human parainfluenza viruses (hPIV 1, 2, 3, and 4), human rhinovirus (hRV), non-rhinovirus enterovirus (EV), and adenovirus (AdV).

## Results

### Study overview

We use detailed individual-level surveillance data from the Seattle Flu Study (SFS), which launched in the Fall of 2018 to improve detection and control of epidemics and pandemics[16]. SFS carried out intensive hospital and community-based surveillance with systematic molecular testing of nasal swabs for up to 26 respiratory pathogens[16] (Table S1). Our study spans November 19, 2018, to June 30, 2022, during which respiratory specimens were collected from individuals with and without respiratory illness across a variety of sites throughout the Seattle metropolitan region, as previously described[16–22]. In total, SFS screened 138,050 respiratory specimens for the presence of 24 or 26 pathogens (Table S1), and we retained 80,891 specimens after limiting our analysis to symptomatic individuals and discarding samples with missing metadata or from multiple testing (Table 1, Table S2). 25.5% (N = 20,659) of samples were collected in hospitals, and 74.5% (N = 60,232) were collected through community-based testing, including outpatient clinics, kiosks stationed in high foot traffic areas[16], swab-and-send at-home testing programs[19,21], and Public Health–Seattle & King County COVID-19 drive through testing sites (Table 1,Table S2). The majority of hospital residuals were collected from younger age groups, while most community-based samples were collected from adults (Table 1, Figure. S1, Table S2).

Over the course of the four-year study, 41% (N = 33,153) of specimens tested positive for at least one respiratory pathogen (including SARS-CoV-2), 32.8% (N = 26,501) were positive for at least one endemic respiratory pathogen, and 9.3% (N = 7,540) were positive for more than one pathogen. Prior to the start of Washington's COVID-19 restrictions in March 2020, the most prevalent pathogens among positive samples were influenza A/H1N1 virus (17.5%), followed by hRV (15.1%), influenza A/H3N2 virus (13.7%), influenza B virus (12%), and RSV A (9.5%) (Figure. S2). After March 2020, the most prevalent pathogens were SARS-CoV-2 (39.3%), hRV (35%), and AdV (5%) (Figure. S2).

We reconstructed daily incidences for SARS-CoV-2 and each endemic pathogen, adjusting for testing volume over time, age, clinical setting, and local syndromic respiratory illness rates (Fig. 1, Figure. S3). Although SFS tested respiratory specimens for up to 26 pathogens, we limited our analysis to 18 viruses with sufficient sampling (see "methods" for inclusion criteria), including SARS-CoV-2, influenza A/H1N1, A/H3N2, and B viruses, RSV A and B, hCoV 229E, OC43, HKU1, and NL63, hPIV 1, 2, 3, and 4, hMPV, hRV, EV, and AdV (Fig. 1, Figure. S3). Due to laboratory assay limitations, we grouped epidemiologically distinct strains into one incidence time series each for hCoV 229E and hCoV OC43 (hereon hCoV 229E + OC43), hCoV HKU1 and hCoV NL63 (hCoV HKU1 + NL63), hPIV 1 and hPIV 2 (hPIV 1 + 2), hPIV 3 and hPIV 4 (hPIV 3 + 4), hRV, EV, and AdV (Table S1).

Based on reconstructed incidences, we used semi-mechanistic epidemiological models to measure the time-varying intensity of transmission via the daily effective reproduction number ($R_t$)[23,24] (Fig. 1). $R_t$ is the average number of secondary cases arising from an infectious individual at time $t$, assuming epidemiological conditions remain identical after that time[25]. To estimate $R_t$ based on dates of infection, we convolved over uncertain incubation period and reporting delay distributions (i.e., delays from symptom onset to

## Table 1 | Participant characteristics

| Variable | All sample sites, N = 80,891[a] | Site Category | | |
| --- | --- | --- | --- | --- |
| | | Hospital residuals, N = 20,659[a] | SFS community surveillance, N = 52,276[a] | COVID-19 drive through testing, N = 7,956[a] |
| **Home residence** | | | | |
| North King County | 47,412 (59%) | 10,464 (51%) | 31,342 (60%) | 5,606 (70%) |
| South King County | 17,964 (22%) | 4,433 (21%) | 11,846 (23%) | 1,685 (21%) |
| Puget Sound, non-King County[b] | 15,515 (19%) | 5,762 (28%) | 9,088 (17%) | 665 (8.4%) |
| **Sex** | | | | |
| Female | 44,691 (55%) | 9,407 (46%) | 30,888 (59%) | 4,396 (56%) |
| Male | 35,795 (44%) | 11,249 (54%) | 21,074 (41%) | 3,472 (44%) |
| Other | 44 ( < 0.1%) | 0 (0%) | 44 ( < 0.1%) | 0 (0%) |
| (Missing) | 361 | 3 | 270 | 88 |
| **Mean (s.d.[c]) age** | 31 (21) | 15 (20) | 35 (18) | 47 (18) |
| **Age group** | | | | |
| <1 | 3,459 (4.3%) | 2,990 (14%) | 469 (0.9%) | 0 (0%) |
| 1-4 | 9,839 (12%) | 6,368 (31%) | 3,341 (6.4%) | 130 (1.6%) |
| 5-17 | 10,698 (13%) | 5,830 (28%) | 4,586 (8.8%) | 282 (3.5%) |
| 18-49 | 40,311 (50%) | 3,310 (16%) | 33,022 (63%) | 3,979 (50%) |
| 50-64 | 10,902 (13%) | 1,209 (5.9%) | 7,515 (14%) | 2,178 (27%) |
| ≥65 | 5,682 (7.0%) | 952 (4.6%) | 3,343 (6.4%) | 1,387 (17%) |
| **Broad age group** | | | | |
| <5 | 13,298 (16%) | 9,358 (45%) | 3,810 (7.3%) | 130 (1.6%) |
| ≥5 | 67,593 (84%) | 11,301 (55%) | 48,466 (93%) | 7,826 (98%) |

[a]n (%).

[b]Pierce, Snohomish, Kitsap, San Juan, Whatcom, Skagit, Island, Clallam, Jefferson, Mason, and Thurston counties.

[c]standard deviation.

Home residence, sex, and age distributions for individuals contributing respiratory specimens to different Seattle Flu Study (SFS) surveillance arms, including hospitals, SFS community testing (e.g., kiosks, swab-and-send at-home testing, outpatient clinics), and Public Health – Seattle & King County COVID-19 drive through testing sites.

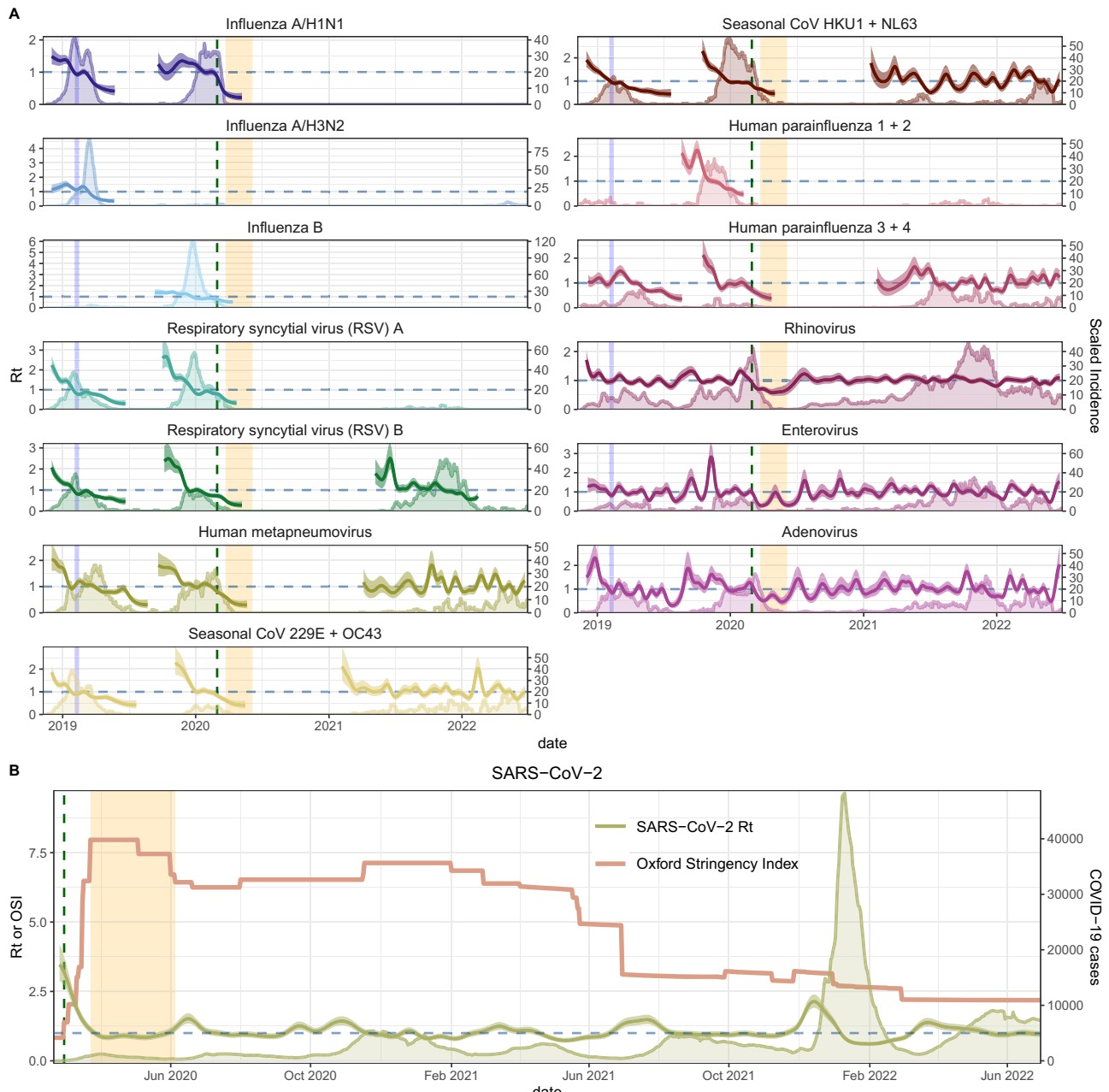

**Fig. 1 | Daily incidence and transmissibility of endemic respiratory viruses and SARS-CoV-2 in the greater Seattle region. A** Daily time-varying effective reproduction numbers ($R_t$, thick lines, left y-axis) and reconstructed incidences of endemic respiratory viruses (thin lines, right y-axis) during November 2018–June 2022. The vertical blue shaded panel indicates the timing of a major snowstorm in Seattle (February 3-15, 2019), the vertical dashed line indicates the date of Washington's State of Emergency declaration (February 29, 2020), and the vertical orange shaded panel indicates Seattle's stay-at-home period (March 23–June 5, 2020). **B** Daily time-varying effective reproduction numbers of SARS-CoV-2 ($R_t$, thick green line, left y-axis), King County COVID-19 case counts (thin green line, right y-axis), and the stringency of non-pharmaceutical interventions in Washington state, measured by the Oxford Stringency Index (thin orange line, left y-axis), during January 2020 – June 2022. In **A** and **B** daily $R_t$ time series show the posterior median (thin dark line) and 90% credible interval (shaded band).

testing), wherein delays were informed by our individual-level surveillance data (*see "Supplementary Methods"*). We used aggregated mobile device location data from SafeGraph and Meta Data for Good to assess the effects of population-level movements on citywide respiratory virus dynamics in pre- and post-pandemic years (Figs. 2–3). During the pandemic period, we also considered the effects of non-mobility behavioral indicators, including the stringency of Washington's government response to COVID-19, measured by the Oxford Stringency Index[26] (Fig. 1), and the proportion of individuals masking in public[27] (Fig. 2).

**Declines in mobility correlate with reduced respiratory virus circulation during a major snowstorm in February 2019**

Most endemic viruses in our study, including influenza A viruses, RSV, hCoV, hPIV 3 + 4, hMPV, hRV, EV, and AdV, circulated during the 2018–2019 winter season. This season was atypical in that Seattle experienced unusually high snowfall during February 2019, prompting widespread school and workplace closures and reduced regional travel from February 3 to February 15, 2019. The mobility categories most impacted by the snowstorm included foot traffic to elementary and high schools, colleges, and transit stations (>75% declines below

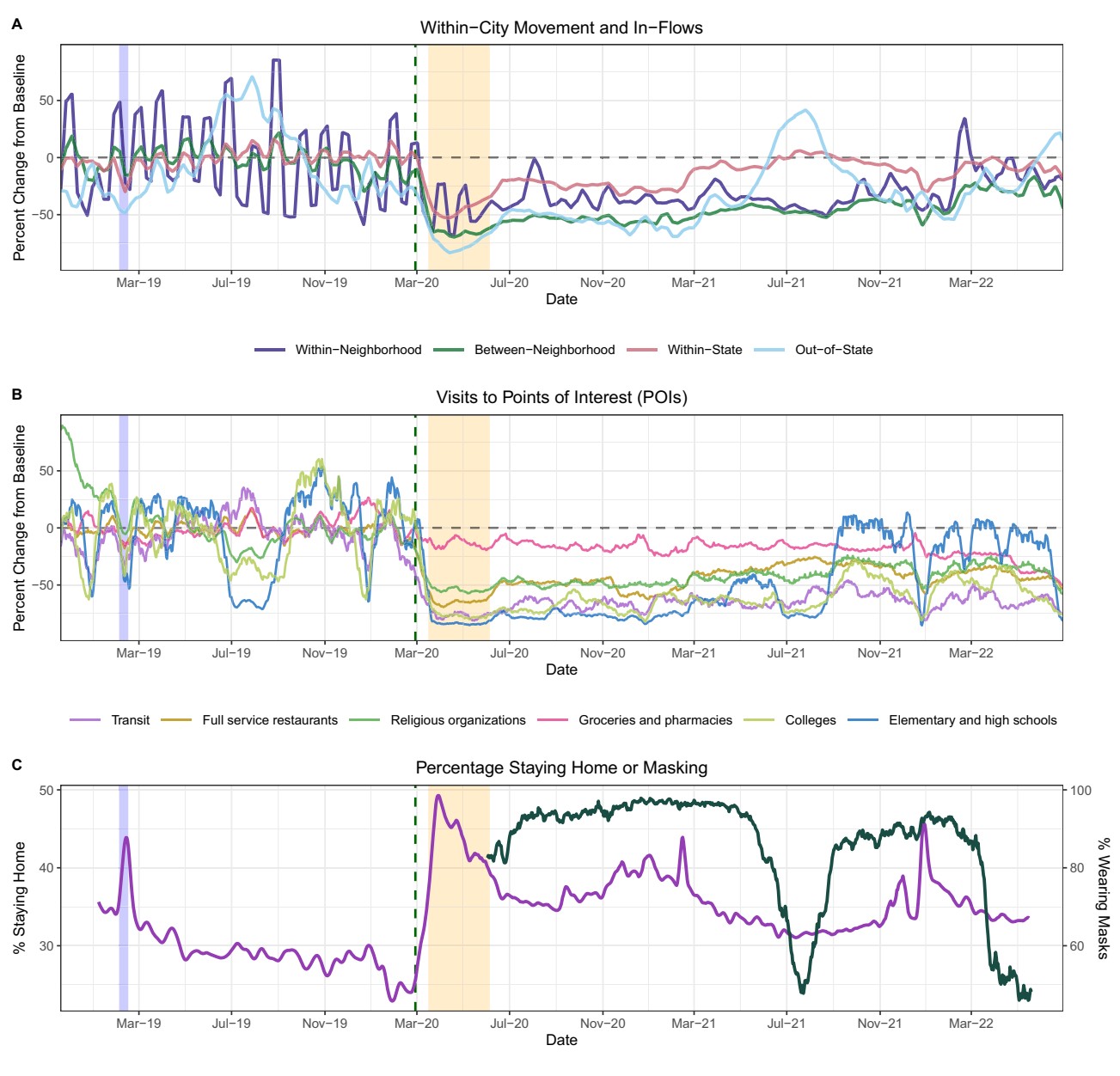

**Fig. 2 | Mobility and behavior trends in the greater Seattle region based on aggregated mobile device location data, November 2018 – June 2022.** In each panel, the vertical blue shaded panel indicates the timing of a major snowstorm in Seattle (February 3–15, 2019), the vertical dashed line indicates the date of the Washington's State of Emergency declaration (February 29, 2020), and the vertical orange shaded panel indicates Seattle's stay-at-home period (March 23–June 5, 2020). **A** The percent change from baseline for large-scale population movements: within-neighborhood movement (purple), between-neighborhood movement (green), inflow of visitors from other Washington counties (red), and inflow of out-of-state visitors (light blue). **B** The percent change from baseline in foot traffic to various categories of points of interest (POIs): transit stations (purple), full-service restaurants (dark yellow), religious organizations (green), groceries and pharmacies (pink), colleges and universities (light green), and elementary and high schools (blue). **C** The percentage of devices staying completely at home (purple, left *y*-axis) and the percentage of individuals masking in public in King County (dark green, right *y*-axis).

baseline), visits to child daycare centers, and the inflow of out-of-state visitors ( > 50% declines below baseline) (Fig. 2, Figure. S4). As previously described[20], this city-wide shutdown led to a conspicuous dip in the incidence of several pathogens (Figure. S5).

To measure the overall impact of the snowstorm on virus circulation, we compared pathogen specific $R_t$ values during the two weeks before and after the start of heavy snowfall on February 3, 2019 (Table 2). RSV, AdV, and EV were the pathogens most affected by weather-related disruptions (33-40% declines), followed by influenza viruses and hCoV (10-20% declines) (Table 2). Influenza A/H3N2 virus,

hPIV 3 + 4, hMPV, hRV, EV, and AdV rebounded after schools and workplaces reopened, and their epidemics subsequently peaked from mid-March to early April 2019 (Figure. S5).

During February 2019, reductions in mobility preceded or coincided with declines in pathogen transmission, though the strength of correlations varied across pathogens (Figures. S6-S7). Among pathogens with the most substantial declines (RSV, AdV, and EV), drops in $R_t$ coincided most closely with reductions in visitor inflow and foot traffic to schools, child day care centers, restaurants, and religious services (mean Spearman's rank cross-correlation

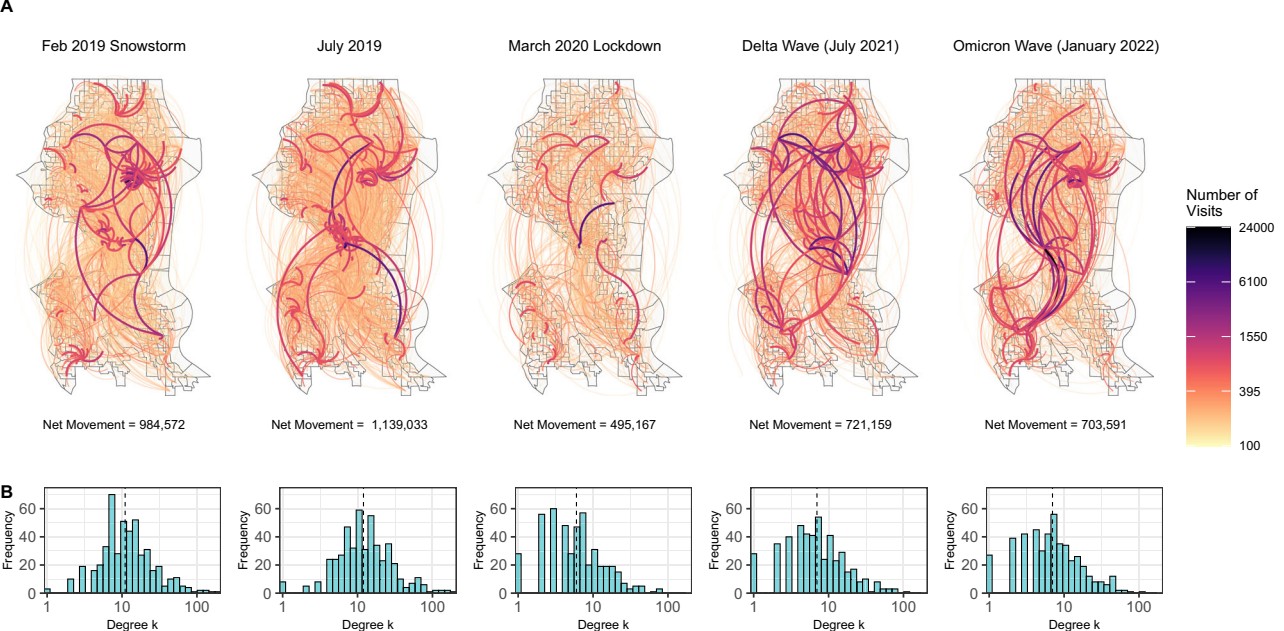

**A**

Feb 2019 Snowstorm | July 2019 | March 2020 Lockdown | Delta Wave (July 2021) | Omicron Wave (January 2022)

Net Movement = 984,572 | Net Movement = 1,139,033 | Net Movement = 495,167 | Net Movement = 721,159 | Net Movement = 703,591

**B**

**Fig. 3 | Undirected network of mobile device movement between neighborhoods (census block groups, CBGs) in Seattle, Washington, during key epidemiological time points.** Time points include a major snowstorm in February 2019, a week in July 2019 to show baseline movement, the beginning of stay-at-home orders in March 2020, a week during the Delta wave in July 2021, and a week during the Omicron BA.1 wave in January 2022. **A** Weekly visitors to points of interest (POI) are aggregated by visitor home CBG and POI CBG. Network edges (lines) are shaded according to the number of unique visitors between each pair of CBGs within a particular week, with thicker, darker edges indicating a greater number of visitors. **B** Histograms showing the frequency of degree $k$ values for Seattle neighborhoods (i.e., the integer number of other neighborhoods to which each individual neighborhood is connected) at each time point. The vertical dashed line overlaying each histogram indicates the median degree of the network of Seattle neighborhoods.

coefficients, $\rho$ range: 0.79 – 0.98; all reported correlations are statistically significant; Figures. S6-S7). For pathogens that did not experience declines in transmission (hPIV 3 + 4, hMPV, and hRV), $R_t$ had negative or non-significant associations with mobility during the snowstorm (Figures. S6-S7).

## Table 2 | Changes in transmissibility (time-varying effective reproduction numbers, $R_t$) during the two weeks before and after two events: a major snowstorm in February 2019 and the initiation of COVID-19 social distancing measures in March 2020

| Pathogen | Major snowstorm, February 2019 | | Early COVID-19 restrictions, March 2020 | |
|---|---|---|---|---|
| | Change in $R_t$ | P-value | Change in $R_t$ | P-value |
| Influenza A/H3N2 | -12 [-17, -7] % | $1 \times 10^{-16}$ | Not circulating | |
| Influenza A/H1N1 | -19 [-24, -15] % | $1 \times 10^{-16}$ | -43 [-54, -33] % | $1 \times 10^{-16}$ |
| Influenza B | Not circulating | | -15 [-17, -12] % | $1 \times 10^{-16}$ |
| RSV A | -40 [-45, -35] % | $1 \times 10^{-16}$ | -9 [-13, -5] % | $1 \times 10^{-16}$ |
| RSV B | -37 [-40, -32] % | $1 \times 10^{-16}$ | -4 [-5, -2] % | $1 \times 10^{-16}$ |
| hMPV | 19 [16, 22] % | 0.001 | -21 [-22, -19] % | $1 \times 10^{-16}$ |
| hPIV 1 + 2 | Not circulating | | Not circulating | |
| hPIV 3 + 4 | 14 [9, 18] % | 0.001 | -21.5 [-22, -21] % | $1 \times 10^{-16}$ |
| hCoV 229E + OC43 | -10 [-14, -5] % | $1 \times 10^{-16}$ | -20 [-21, -19] % | $1 \times 10^{-16}$ |
| hCoV HKU1 + NL63 | -18 [-19, -17] % | $1 \times 10^{-16}$ | -25 [-28, -22] % | $1 \times 10^{-16}$ |
| hRV | 2 [0, 3] % | 0.073 | -29 [-32, -26] % | $1 \times 10^{-16}$ |
| EV | -33 [-40, -25] % | $1 \times 10^{-16}$ | -31 [-53, -13] % | 0.003 |
| AdV | -39 [-43, -33] % | $1 \times 10^{-16}$ | -33 [-45, -21] % | $1 \times 10^{-16}$ |
| SARS-CoV-2 | Not circulating | | -39 [-54, -25] % | $1 \times 10^{-16}$ |

We compared $R_t$ values before and after each event using nonparametric bootstrap tests for the ratio of two means. Mean ratios, 95% confidence intervals, and two-sided p-values were estimated from 1000 resamples.

## Relationships between mobility and pathogen transmission during the 2019-2020 winter season (pre-pandemic)

The 2019-2020 virus respiratory season was a relatively typical season in Seattle with heightened activity of many common respiratory viruses (Fig. 1). During Fall 2019, visits to child daycares, schools, colleges, and religious organizations preceded or coincided with initial increases in the circulation of influenza viruses, RSV, hMPV, hCoV, AdV, and hPIV (moving window Spearman's rank cross-correlation coefficients, $\rho$ range: 0.49 – 0.93; all reported correlations are statistically significant; Fig. 4, Figure. S8). Large-scale movements also correlated with changes in endemic virus transmissibility, in particular, the percentage of devices leaving home (RSV, hCoV HKU1 + NL63, hPIV 3 + 4, and influenza A/H1N1; $\rho$: 0.5 – 0.82) and between-neighborhood movement (AdV, hMPV, RSV A, hPIV 1 + 2, and influenza B; $\rho$: 0.48 – 0.82) (Fig. 4, Figure. S8). For most pathogens, the strongest relationships between transmission and mobility occurred at the beginning of the season in early autumn (Fig. 4, Figure. S8).

We used multivariable generalized additive models (GAMs) to measure non-linear relationships between mobility and $R_t$ and model selection of GAMs to assess the relative importance of different indicators in predicting $R_t$ during the 2019–2020 season, prior to the start of the COVID-19 pandemic (September 2019–February 2020). For each pathogen, we allowed candidate models to include a smoothed temporal trend and up to two smoothed mobility terms. GAMs were fit to timeframes spanning the exponential growth phase of each outbreak, when $R_t$ exceeds 1 and susceptible depletion is limited. For most pathogens, minimal models included a school-related behavioral indicator (foot traffic to schools or colleges) and a covariate related to large-scale population movement (the percentage of devices leaving home or out-of-state inflow), with the partial effects of most mobility covariates monotonically increasing with $R_t$ (Figure. S9).

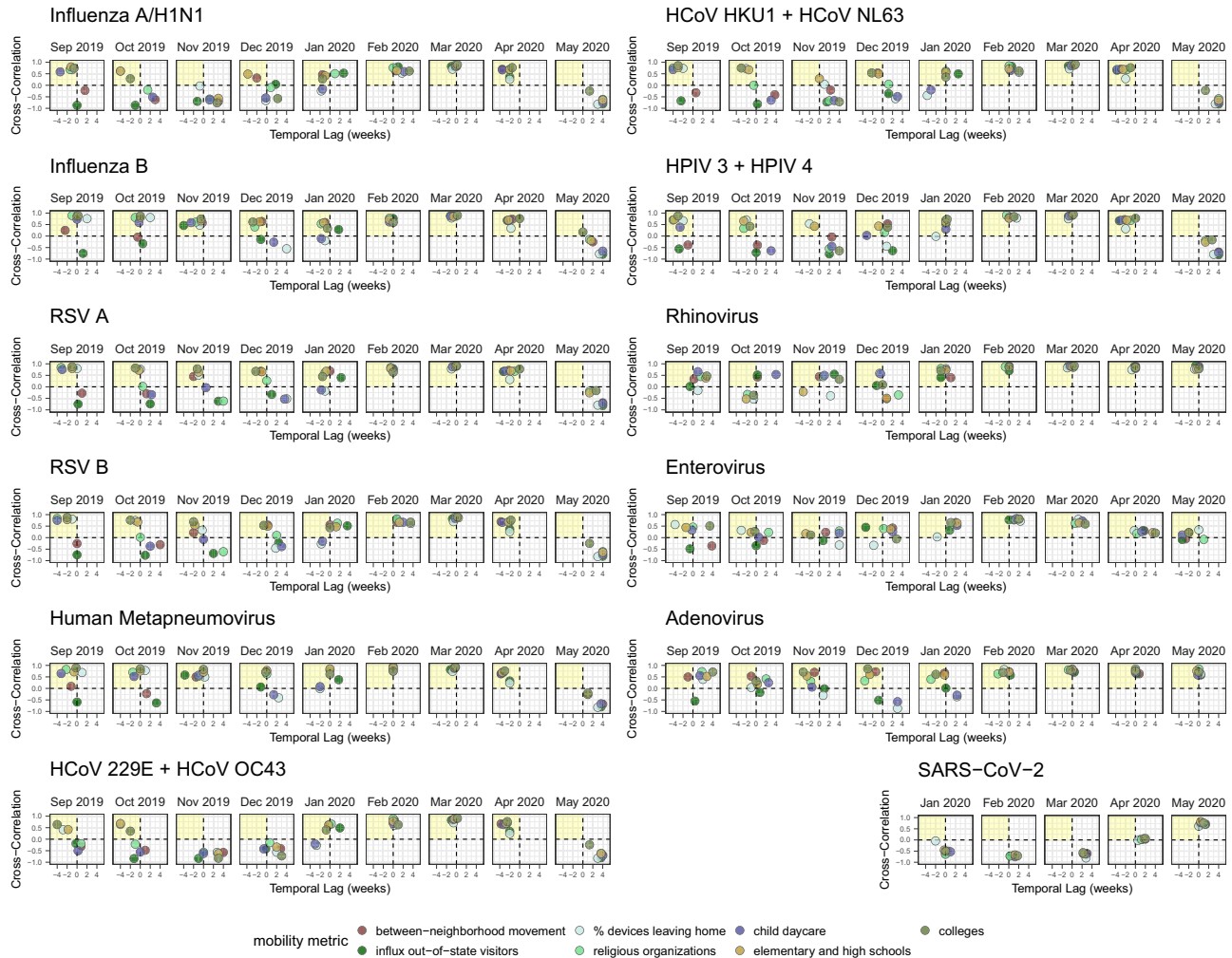

**Fig. 4 | Time series cross-correlations and optimal lags between respiratory virus transmissibility (time-varying effective reproduction numbers, $R_t$) and cell phone mobility in the greater Seattle region, September 2019 – May 2020.** Points are individual mobility indicators derived from aggregated mobile device location data. Spearman's rank correlation coefficients are on the y-axis, and temporal lags (in weeks) between $R_t$ and mobility are on the x-axis. Negative lags indicate behavior leads $R_t$, and positive lags indicate $R_t$ leads behavior. A lag of 0 indicates the two time series are in phase (i.e., synchronous). The yellow shaded panel in each facet includes mobility indicators that have a leading, positive relationship with transmission, and hence would be considered predictive of transmission. Figure. S10 shows cross-correlations for the full set of mobility indicators.

## Initial effects of COVID-19 restrictions on mobility and respiratory virus circulation

The first SARS-CoV-2 infections in Washington state arose from a single introduction in late January or early February 2020, and at least one clade was circulating in the Seattle area for 3–6 weeks prior to February 28, when the first community-acquired case was reported[17]. To slow the spread of SARS-CoV-2, Washington declared a State of Emergency on February 29, closed schools in King, Pierce, and Snohomish counties on March 12, and enacted statewide stay-at-home (SAH) orders on March 23. In the interim, King County recommended that workplaces allow employees to work from home on March 4 and closed indoor dining and many other businesses on March 16.

Mobility levels declined substantially after February 29, and by the start of King County's business closures on March 16, foot traffic to transit stations was > 90% below baseline, foot traffic to schools and colleges was > 80% below baseline, and out-of-state inflow and within-city mixing were > 60% below baseline (Fig. 2, Figure. S4). After the enactment of SAH orders on March 23, foot traffic to POIs and large-scale movements declined to 65-95% below baseline (Fig. 2, Figure. S4), while the percentage of devices staying completely at home increased to 50% (Fig. 2). Notably, social distancing measures altered not only the volume of movement between Seattle neighborhoods but also the presence and absence of connections between neighborhoods (Fig. 3).

The transmission rates of all respiratory pathogens dropped substantively after the State of Emergency, though some seasonal pathogens were already declining prior to February 29 (Fig. 1, Figure. S3). We measured the initial impacts of COVID-19 NPIs on respiratory virus circulation by comparing $R_t$ values during the 2 weeks before and after the State of Emergency declaration on February 29 (Table 2). Early public health measures were effective at lowering SARS-CoV-2 transmission rates by 39% [95% CI: 25, 54]. Among endemic pathogens, influenza A/H1N1 virus, AdV, EV, and hRV were the most impacted by pandemic-related behavioral changes, experiencing 29 – 43% reductions in transmissibility by March 16, followed by hPIV 3 + 4, hMPV, and hCoV (20 – 25% declines). Reductions in RSV and influenza B virus were more modest, given their seasonal outbreaks had mostly concluded by late February. The hPIV 1 + 2 outbreak subsided in mid-February and thus was not impacted by COVID-19 NPIs.

We observed strong relationships between mobility and the transmission of respiratory pathogens in the Spring of 2020. All mobility metrics were positive, leading indicators of $R_t$ across all

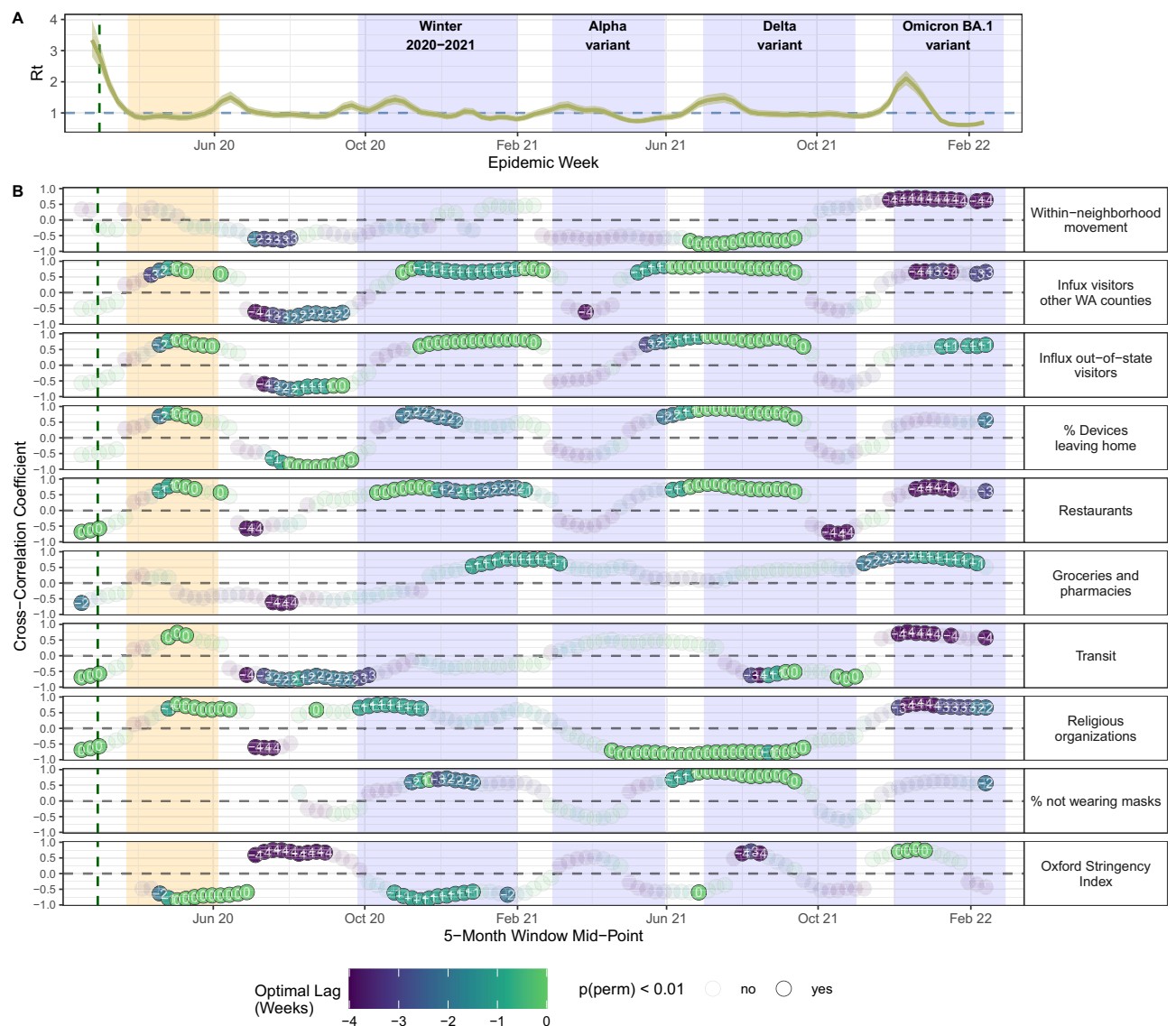

**Fig. 5 | Relationships between SARS-CoV-2 transmission and mobility, masking, and the stringency of non-pharmaceutical interventions in the greater Seattle region, January 2020 – March 2022.** The vertical green dashed line indicates the date of Washington's State of Emergency declaration (February 29, 2020), and the vertical orange shaded panel indicates Seattle's stay-at-home period (March 23 – June 5, 2020). The blue shaded panels indicate the timing of four COVID-19 waves in Seattle. **A.** Weekly effective reproduction numbers ($R_t$) of SARS-CoV-2, including the posterior median (thin dark line) and 90% credible interval (shaded band). **B.** Rolling Spearman's rank cross-correlations between weekly $R_t$ and behavioral indicators. Points represent the maximum (absolute) coefficient values for 5-month rolling cross-correlations between weekly $R_t$ and individual mobility metrics, after constraining the analysis to leading or synchronous relationships between mobility and $R_t$. Point color and the number within each point indicate the lag in weeks corresponding to the maximum cross-correlation coefficient value for each 5-month period ("optimal lag"). Negative values indicate that behavior leads $R_t$, and a lag of 0 indicates the time series are in phase (i.e., synchronous). Point transparency indicates statistical significance of the cross-correlation coefficient (yes: solid, no: transparent).

endemic viruses (Fig. 4, Figures. S10-S11). In contrast, mobility lagged and was negatively correlated with SARS-CoV-2 transmission during this period (Fig. 4, Figures. S10-S11). COVID-19 restrictions began to relax on May 4, 2020, when King County entered Phase 1 of the state's reopening plan, allowing outdoor dining, worship services, and fitness centers to reopen with limited capacity. SARS-CoV-2 $R_t$ values ranged from 0.8 to 0.9 throughout April and May and did not surpass 1 until early June (Fig. 1). During late April and May, SARS-CoV-2 $R_t$ became positively correlated and synchronous with most mobility indicators (moving window Spearman's $\rho$: 0.59 – 0.92) and inversely correlated with the stringency of NPIs ($\rho$: -0.85 – -0.7) (Fig. 4, Figures. S10-S11). Yet, these relationships did not persist after the virus's initial rebound in early June 2020, when King County reopened indoor dining and additional businesses (Figures. S10-S11).

Due to simultaneous changes in all mobility metrics, we could not differentiate the effects of individual indicators on $R_t$ during Seattle's SAH period.

Population mobility did not immediately recover after SAH orders were lifted in June 2020 (Figs. 2-3, Figure. S4). Visitor inflow from other WA counties and US states remained depressed at levels 50% below baseline until the spring and summer months of 2021, and within-and-between-neighborhood movement had not returned to pre-pandemic levels by the conclusion of our study in June 2022 (Figs. 2-3, Figure. S4). Further, SAH orders caused long-lasting structural changes to the mobility network of Seattle, wherein neighborhoods with a high degree of centrality ("hubs") were most affected (Fig. 3). After March 2020, neighborhoods with fewer than 10 connections to other neighborhoods became much more prevalent in the network, causing an

overall downshift in the mobility network's median degree for the remainder of the study period (Fig. 3).

## Associations between SARS-CoV-2 transmission and behavior differ across COVID-19 waves

We measured cross-correlations between SARS-CoV-2 transmission, mobility, masking, and NPI stringency during subsequent pandemic waves in Seattle (Fig. 5). After its first two COVID-19 outbreaks in Spring 2020 and Summer 2020, Seattle experienced a large third wave during winter 2020-2021 despite high masking rates (Fig. 5). During the early, exponential growth stage of the outbreak, increases in $R_t$ coincided with foot traffic to religious services and restaurants (moving window Spearman's $\rho$: 0.57 – 0.76). During the remainder of the wave, the percentage of devices leaving home ($\rho$: 0.57 – 0.85), influx of visitors from other WA counties and US states ($\rho$: 0.65 – 0.83), and stringency of NPIs ($\rho$: -0.85 – -0.61) preceded or coincided with dynamics in $R_t$ (Fig. 5). A smaller wave associated with the Alpha variant spanned March to May 2021, during which $R_t$ was not significantly associated with mobility (Fig. 5). COVID-19 cases and hospitalizations surged again in July 2021, due to the spread of the highly transmissible Delta variant. From June to late July 2021, increases in $R_t$ were synchronous with foot traffic to restaurants, the percentage of devices leaving home, visitor inflow, and the proportion of individuals not masking in public ($\rho$: 0.66 – 0.94) (Fig. 5). The Omicron BA.1 variant was first identified in the U.S. on December 1, 2021, and confirmed in Washington state on December 4, 2021. Compared to prior variants of concern (VOCs), Omicron BA.1 had increased immune evasion and greater intrinsic transmissibility[28,29]. In Seattle, slight increases in within-neighborhood movement and foot traffic to groceries and pharmacies preceded the rapid uptick in Omicron cases during December 2021 ($\rho$: 0.63 – 0.89), while drops in several mobility indices led the sharp decline in cases during January 2022 (Fig. 5). Minimal GAMs fit to the exponential growth phase of each COVID-19 wave retained the percentage of devices leaving home, the proportion of individuals masking in public, foot traffic to restaurants, or out-of-state inflow (Figure. S12), though associations between behavior and $R_t$ were often nonlinear.

## Mobility predictors of endemic pathogen rebound during the COVID-19 pandemic

We observed a remarkably fast rebound of hRV, and AdV and EV to a lesser degree, when SAH restrictions relaxed in early June 2020 (Fig. 1). Increases in hRV transmission were preceded by or synchronous with foot traffic to child daycares, restaurants, and transit stations and large-scale movements, including the percentage of devices leaving home, within-city mixing, and out-of-state inflow, from June to early August 2020 (moving window Spearman's $\rho$: 0.54–0.96; Figure. S13), inversely correlated with NPI stringency from late August to early October 2020 ($\rho$: -0.82 to -0.58), and continuously correlated with foot traffic to religious organizations and inflow from other WA counties until early 2021 ($\rho$: 0.55–0.97) (Figure. S13). From June to late August 2020, the rebound of AdV was preceded by or synchronous with slight increases in foot traffic to schools and religious organizations ($\rho$: 0.55–0.88) and negatively correlated with NPI stringency ($\rho$: -0.74 to -0.61) (Figure. S13). Lastly, EV $R_t$ briefly correlated with most mobility metrics during its initial rebound in June and July 2020 ($\rho$: 0.53–0.81; Figure. S13). For all three viruses, minimal GAMs fit to the first six months of rebound retained foot traffic to religious services and out-of-state inflow (Figure. S14).

Resurgence of other endemic viruses, including hCoV, hPIV, hMPV, and RSV B, was not observed until early-to-mid 2021, and epidemic peaks for hCoV, hMPV, and RSV B occurred during the spring or summer outside of their typical seasons (Fig. 1). We measured univariate associations between mobility, masking, NPI stringency, and daily transmissibility and found fewer clear relationships compared to

Seattle's SAH period and the 2019-2020 winter season. Several mobility indicators preceded or coincided with the rebound of these viruses (Figure. S15). For example, at the beginning of RSV B's rebound in summer 2021, increases in $R_t$ were preceded by between-neighborhood movement and foot traffic to schools, child daycares, and religious services ($\rho$: 0.63 – 0.88) and synchronous with the percentage of devices leaving home, the proportion of individuals not masking in public, foot traffic to restaurants, and out-of-state inflow ($\rho$: 0.57 – 0.86) (Figure. S15). However, associations between mobility and RSV B $R_t$ were more transient compared to the pre-pandemic period, persisting for 2-4 months (Figure. S15) instead of 3-6 months (Figure. S8). Minimal GAMs fit to the first months of each virus's rebound retained between neighborhood movement, out-of-state inflow, or foot traffic to religious services, though relationships between mobility and $R_t$ were often nonlinear and not consistently positive (Figure. S16).

In late 2021, endemic virus circulation declined as Omicron cases surged, and reductions in endemic virus transmission were preceded by or coincided with drops in mobility (Figures. S13, S15, S17). For example, reductions in most mobility indicators preceded declines in RSV B, hMPV, hCoV, AdV, and hPIV 3 + 4 circulation by 1 to 4 weeks, while reduced visits to child daycares, religious services, restaurants, colleges, and transit stations were synchronous with declines in hRV transmission (Figures. S13, S15). The transmission rates of RSV B, hMPV, hCoV, RV, and AdV also correlated with the percentage of devices staying home, which spiked from 30% to 50% in December 2021 (Fig. 2, Figure. S17). During the Omicron BA.1 wave, the best fit minimal GAMs for endemic viruses retained a school-related behavioral indicator, the percentage of devices leaving home, between-neighborhood movement, or out-of-state inflow, similar to results observed for the 2019-2020 season (Figure. S18).

## Short-term forecasting of daily transmissibility using mobility data

Although our study's aim was inferential rather than predictive, we built forecasting models predicting daily $R_t$ at one-week horizons for three viruses that circulated continuously throughout the study period: hRV, AdV, and SARS-CoV-2 (Figures. S19-S21; see "Supplementary Methods"). We evaluated the predictive power of cellphone-derived mobility metrics, the co-circulation of other viruses, and climatic variables, in combination with the past activity of the target virus during the previous two weeks (14 autoregressive terms), against a baseline model that only included autoregressive (AR) terms. An additional model for SARS-CoV-2 spanning 2021-2022 included covariates for vaccination coverage and variant emergence (Figure. S22).

For each virus, models including mobility and AR terms produced generally accurate forecasts over the entire study period (RMSE: hRV: 0.013; AdV: 0.04; SARS-CoV-2: 0.03; Table S3), and especially during Seattle's stay-at-home orders and the initial lifting of restrictions (RMSE: hRV: 0.006; AdV: 0.02; SARS-CoV-2: 0.02; Table S4). When forecasting SARS-CoV-2 $R_t$ during 2021-2022, covariates for vaccination and variant circulation did not improve model accuracy, but models including mobility had a 13% improvement in prediction RMSE relative to the baseline model (Table S5). When assessing model accuracy over the entire study period, models including mobility, climatic variables, or viral interference did not outperform baseline models for any of the three viruses (Figures. S19-S21; Table S3). Thus, although mobility data can provide small to moderate benefits to prediction accuracy, the additional information provided by past population movements is limited in comparison to the knowledge of past disease incidence[30].

## Discussion

We investigated the impacts of human behavior on the transmission of respiratory viruses in the greater Seattle region during pre- and post-

pandemic years by modeling incidence derived from hospital and community-based respiratory surveillance and human movements from high-resolution mobile device location data. From November 2018 to June 2022, we characterized the epidemiological dynamics of 17 endemic viruses and SARS-CoV-2 and related changes in daily transmissibility (time-varying effective reproduction numbers, $R_t$) to trends in population mobility, masking, and COVID-19 non-pharmaceutical interventions (NPIs). To our knowledge, this is the first study to explore the effects of mobility and behavior on transmission across a large set of endemic pathogens; interestingly, we saw notable heterogeneity in the timing and size of each endemic pathogen's rebound during the pandemic period.

Mobility was most predictive of transmission during periods of dramatic behavioral change, as observed during Seattle's stay-at-home (SAH) orders in March 2020. Smaller-scale changes in mobility also preceded or coincided with increases in $R_t$ at the beginning of outbreaks and with declines in $R_t$ during shorter interruptions to human movement, as observed during a major snowstorm in February 2019 and the Omicron BA.1 wave in late 2021. Across all endemic viruses, trends in daily $R_t$ were repeatedly associated with the same set of mobility metrics, including foot traffic to elementary and high schools, colleges, child daycare centers, and religious organizations, the percentage of devices leaving home, between-neighborhood movement, and the inflow of visitors from other WA counties and US states. After the SAH period, SARS-CoV-2 transmission correlated with foot traffic to restaurants, religious organizations, and colleges, the percentage of devices leaving home, and visitor inflow. Foot traffic to specific businesses and educational and religious activities may approximate close contacts or crowded conditions that facilitate direct, aerosol, or droplet transmission, while the percentage of devices leaving home, within-city mixing, and inflow from other regions may be indicative of human movements that promote viral introductions and dispersal.

The age distribution of infections may explain slight differences in which categories of POIs correlated with endemic virus transmission versus SARS-CoV-2 transmission. Recurrent associations between endemic virus $R_t$ and visits to schools and daycares are consistent with children experiencing the highest rates of (endemic) respiratory infections and schools and daycares acting as a major source of transmission in the community[31–34]. Unlike endemic respiratory viruses, all age groups are susceptible to SARS-CoV-2 infection. Correlations between SARS-CoV-2 $R_t$ and foot traffic to colleges, religious organizations, and restaurants, but not schools or daycares, could be attributed to greater rates of symptomatic infection (and hence shedding propensity) in adults relative to younger age groups[35] or to the greater relevance of adult networks in spreading SARS-CoV-2 compared to endemic viruses.

COVID-19 NPIs significantly perturbed the transmission of respiratory pathogens at a global level[1–8], causing the complete disappearance, delayed return, or "off-season" outbreaks of endemic pathogens[6–9]. In Seattle, all endemic respiratory viruses experienced rapid declines at the beginning of Seattle's SAH orders in March 2020, but as restrictions eased, their rebound was heterogeneous. Similar to trends observed in the US and other countries[4,6,7,10,36,37], the circulation of hRV, EV, and AdV resumed in early June 2020, immediately after nonessential businesses and indoor dining reopened, while other respiratory viruses virtually disappeared during March 2020 and did not recirculate until 2021. Further, the resurgence of RSV B, hCoV, and hMPV occurred outside of their typical seasons, as reported in other locations[7–9]. After the initial easing of COVID-19 restrictions, relationships between endemic virus dynamics and mobility were less clear compared to Seattle's SAH orders or the 2019-2020 winter season, potentially due to continued social distancing and masking, a more refined understanding of "high-risk" activities, the delay of in-person instruction for school students until spring 2021, or structural changes to Seattle's mobility network. Nonetheless, associations between

endemic virus $R_t$ and population behavior were overall stronger and longer-lasting than those observed for SARS-CoV-2.

It is remarkable that the three viruses that rebounded immediately after lockdown restrictions lifted, hRV, EV, and AdV, are non-enveloped viruses, while the other viruses studied here are enveloped. The immediate rebound of non-enveloped viruses could be attributed to viral stability and persistence. Non-enveloped viruses are less susceptible to lipophilic disinfection and can persist on hands and fomites for longer periods of time than enveloped viruses[38,39]. In addition to the presence or absence of an envelope, several other factors, such as transmission mode, seasonality, source/sink dynamics, and duration of infectious period and immunity, could have affected the timing of rebound. While enveloped viruses disappeared in March 2020, non-enveloped viruses may have continued to spread during SAH restrictions, due to their longer periods of viral shedding, high preexisting community prevalence, or ability to persist on environmental surfaces[37–40]. We hypothesize that low levels of transmission or residual viral particles on surfaces, combined with slight increases in movement, close contacts, and visitor inflow, were sufficient to facilitate the rapid rebound and ongoing transmission of hRV, EV, and AdV after SAH orders were lifted in June 2020. Further, surgical masks are less effective at filtering hRV compared to influenza viruses and seasonal coronaviruses[41].

Our study period encompasses the two respiratory virus seasons prior to the start of the COVID-19 pandemic and two pandemic years. The Seattle Flu Study (SFS) began collecting samples in November 2018, which precluded us from evaluating potential leading indicators of transmission at the beginning of the 2018-2019 season. However, we were able to detect strong links between mobility and transmission in February 2019 when a major snowstorm forced work and school closures, consistent with previous SFS research that did not specifically examine cell phone mobility patterns[20]. SFS continued to collect respiratory samples throughout 2019, enabling us to test for leading indicators of transmission during the 2019-2020 winter season. During Fall 2019, the transmission dynamics of enveloped viruses were more strongly correlated with mobility than those of non-enveloped viruses. For enveloped viruses, foot traffic to schools and colleges, between-neighborhood movement, and visitor inflow preceded or coincided with increases in transmission, with associations between $R_t$ and mobility weakening over the course of the season, presumably due to accumulating immunity in the population. During this same period, non-enveloped viruses had fewer positive relationships with mobility, potentially because hRV, EV, and AdV circulate year-round and have less defined peaks and troughs.

SARS-CoV-2 began circulating in the greater Seattle region during January or February 2020[17], with the first community-acquired case confirmed on February 28, 2020. Mobility had a negative, lagging relationship with SARS-CoV-2 $R_t$ during the early months of 2020, suggesting that Seattle residents adjusted their behavior in response to COVID-19 case counts or restrictions rather than the reverse. This result is consistent with a comprehensive analysis of US counties, which found that not all major cities (e.g., San Francisco) experienced strong positive associations between mobility and infection growth rates during the first wave[42]. A caveat is that $R_t$ estimates during this time were likely upwardly biased due to low case counts, sampling biases, and rapid increases in testing capacity and may not have fully captured the steepness of transmission declines after NPIs were implemented. To mitigate this issue, we limited SARS-CoV-2 $R_t$ estimates to dates after February 28, 2020, when the first community-acquired case was detected and cumulative confirmed cases exceeded 50. We also applied centered smoothing windows to case counts, $R_t$ estimates, and mobility indicators so that cross-correlations captured broad signals that were accurately oriented in time[43,44]. After March 2020, mobility was briefly predictive of SARS-CoV-2 transmission when social distancing restrictions first relaxed in summer 2020 and during

the winter 2020–2021, Delta and Omicron BA.1 waves, though relationships were less clear compared to the stay-at-home period.

Climate affects the stability and seasonal dynamics of respiratory viruses[45,46] but was an unlikely driver of endemic virus resurgence in Seattle. hRV, EV, and AdV have year-round circulation with peaks in the spring and late summer or early autumn[47], while influenza viruses, RSV, hCoV, hMPV, and hPIV have distinct seasonality with peaks during the winter or spring[48,49]. The lifting of SAH orders in June 2020 coincided with the typical timing of low circulation for enveloped viruses and increasing activity for non-enveloped viruses. However, the intersection of relaxing NPIs with warmer weather cannot account for the global differences observed between non-enveloped and enveloped virus rebound. The prolonged absence of influenza and RSV circulation was also reported during the Southern Hemisphere winter[2,5,9], and climatic factors cannot explain the rebound of hCoV, hMPV, and RSV B outside of their typical seasons.

Our findings suggest that in-person school instruction played a key role in the rebound of enveloped viruses in Seattle. Prior research has shown that increased contact rates among older children during school terms influence the timing of seasonal influenza and "common cold" virus outbreaks[31,50,51], and younger children and adults acquire influenza and RSV infections from preschool or school-aged children living in the same household[52,53]. All King County public school districts began the 2020-2021 academic year remotely[54], with some districts offering limited in-person instruction to special education students during Fall 2020. Foot traffic to schools was 75% below baseline in Fall 2020, gradually increased to 50% below baseline during Spring 2021, and returned to baseline levels in Fall 2021. We observed that the circulation of hCoV, hPIV, and hMPV increased after elementary school students were offered in-person instruction in February and March 2021 and that the off-season RSV B wave in summer 2021 began directly after hybrid learning became available to all grades in mid-April[54]. These trends suggest that a year of remote learning and, in turn, reduced contacts among school-aged children contributed to the delayed return of enveloped virus circulation to Seattle.

The rebound of enveloped viruses also coincided with increasing rates of travel into Seattle. Annual influenza epidemics in North America are seeded via air travel by strains originating in East and Southeast Asia[55], and the regional spread of influenza viruses correlates closely with work commutes[56]. We did not have data on international air travel or commuting patterns, but cell phone data show that the inflow of visitors from other WA counties and US states was 50% below baseline throughout 2020 and did not return to pre-pandemic levels until late spring or summer 2021. Although the contribution of local persistence versus external seeding is less understood for other seasonal respiratory viruses, increasing inflow into Seattle likely imported cases from other regions, seeding new outbreaks[9].

Lastly, prolonged lack of exposure due to reduced viral circulation during 2020 and 2021 is expected to have increased the cohort of children completely naïve to various respiratory viruses and the waning of immunity in previously infected individuals[12,14]. This "immunity debt" may have provided enough susceptible individuals to sustain spring and summer outbreaks of enveloped viruses. Although we expected these outbreaks to be larger or more severe than those observed during pre-pandemic seasons, substantial influenza and RSV epidemics did not occur until the Fall of 2022, potentially due to Seattle residents continuing to social distance and mask throughout 2021 or negative interference between Omicron BA.1 viruses and endemic viruses during the 2021-2022 winter season. After the conclusion of our study, the 2022-2023 season saw atypically early outbreaks of influenza and RSV and higher hospitalization rates in children and adolescents compared to pre-pandemic seasons[13,57].

This study has limitations related to the type of cell phone mobility data used, its geographic scope, and the underlying demographics of mobile device data in general. Young children experience the highest rates of endemic respiratory infections, but SafeGraph does not track individuals younger than 16 years of age. Nonetheless, we found that visits to schools and daycares were positive, leading indicators of transmission, both prior to and during the pandemic, demonstrating that cell phone data collected from adults can approximate the movements or contacts of children. Second, relationships between cellphone mobility and transmission are weaker in more sparsely populated areas[30,42,58], due to differences in the data generation process and representativeness between urban and rural locations. Because our study is limited to a single metropolitan region, our findings may not be applicable to rural counties. Third, although we found statistically significant associations between aggregate movement patterns and pathogen transmission, spatial colocation of mobile devices may better approximate the interpersonal contacts that underlie transmission[30,59]. Although we would have liked to incorporate a spatial colocation metric into our analysis, at the time of writing, Meta Data for Good's Colocation Map dataset was discontinued for our study period, and we did not know of other spatial location datasets that are publicly accessible. Lastly, longitudinal cross-sectional surveys on social interactions, such as the CoMix survey in England, can provide more direct measures of epidemiologically relevant behavior and more representative samples of populations than mobile device data[60]. However, to our knowledge, similar data do not exist for the US.

Our findings are subject to other limitations. First, due to the limited number of seasons in our study, we could not determine if leading indicators of transmission are consistent across timespans longer than four years. Although SFS continued to conduct respiratory surveillance into the 2022-2023 winter season, its community surveillance approach changed substantially after July 2022, making it difficult to extend our study. Second, variability in test volume over time caused SFS surveillance to sometimes miss less prevalent pathogens. For example, SFS detected only a few cases during a small influenza A/H3N2 wave in spring 2022. Third, our multiplex PCR assay could not distinguish between types, strains, or serotypes of some pathogens (hCoV, hPIV, AdV, EV, and hRV). Consequently, our $R_t$ estimates may average over heterogeneities in transmission dynamics among viruses belonging to the same species[18]. Fourth, previous work has shown that SARS-CoV-2 transmission dynamics differed between North and South King County[19,61], presumably due to socioeconomic inequities (e.g., differences in income, household size, and proportion of essential workers) and North King County maintaining a greater reduction in mobility over time. However, we did not have sufficient surveillance data to explore geographic differences in the transmission dynamics of endemic viruses. Fifth, population immunity may modulate relationships between mobility and transmission; analysis of serologic data could shed light on this question. Additional research is needed to delineate the contributions of an increasingly susceptible population and decreased social distancing to the rebound of endemic viruses.

In summary, mobility patterns are most predictive of respiratory virus transmission during drastic changes in contacts and, to a lesser extent, at the beginning of epidemic waves. During the pandemic period, endemic respiratory viruses exhibited stronger relationships with mobility than pandemic SARS-CoV-2. As SARS-CoV-2 transitions to endemicity, relationships with mobility could gradually start to operate similarly to those of other enveloped viruses. Our study shows that mobile phone data can approximate transmission-relevant contacts and has the potential to support the surveillance of endemic respiratory viruses, with the caveat that relationships between transmission and mobility vary depending on the pathogen, magnitude of mobility change, and phase of the epidemic[30,42]. Future research should consider other host factors, such as prior immunity, and more direct proxies of interpersonal contacts to further disentangle

relationships between population behavior and respiratory virus dynamics.

## Methods

### Virologic surveillance and laboratory methods

This population-level study uses cross-sectional surveillance data collected through the Seattle Flu Study (SFS) from November 2018 to June 2022. Initiated in November 2018, SFS was a multi-arm surveillance study of influenza and other respiratory pathogens in the greater Seattle, Washington region, that utilized community and hospital-based sampling[16]. In its first 1.5 years, SFS tracked the transmission of influenza and other respiratory pathogens in the Seattle region by testing swabs collected at hospitals, community sites (e.g., kiosks in high foot traffic areas, outpatient clinics, workplaces, college campuses), and through its swab-and-send at-home testing study[16,21] (Table S2). The SFS team launched the greater Seattle Coronavirus Assessment Network (SCAN) in March 2020 to detect and understand the spread of SARS-CoV-2[22]. SCAN deployed self-administered at-home testing kits to monitor the spread of both SARS-CoV-2 and endemic respiratory pathogens from March 2020 to July 2022. Protected Health Information (PHI) from study participants and patients contributing residual samples were collected through REDCap projects built by the Institute of Translational Health Sciences (ITHS). We describe each surveillance arm in the Supplementary Methods.

Each respiratory specimen was screened in duplicate for a panel of respiratory pathogens using a custom TaqMan RT-PCR OpenArray panel (Thermo Fisher)[62]. Laboratory methods are described in detail elsewhere[16,19,21]. Pathogen targets included adenovirus (AdV); human bocavirus (hBoV); human coronaviruses (hCoV) 229E, OC43, HKU1, and NL63; human metapneumovirus (hMPV); human parainfluenza viruses (hPIV) 1, 2, 3, and 4; human parechovirus (hPeV); influenza A (IAV) H1N1 and H3N2; pan influenza A (IAV); pan influenza B (IBV); pan influenza C (ICV); respiratory syncytial viruses (RSV) A and B; human rhinovirus (hRV); enterovirus D68 (EV.D68); pan enterovirus excluding D68 (EV); measles; mumps; *Streptococcus pneumoniae* (SPn); *Mycoplasma pneumoniae* (MPn); *Chlamydia pneumoniae* (CPn); and SARS-CoV-2 (Table S1). For a specimen to be designated positive for a given target, both duplicates must test positive in replicate RT-PCR assays. Due to assay limitations, epidemiologically distinct strains were grouped into one assay each for hCoV 229E and hCoV OC43, hCoV HKU1 and hCoV NL63, hPIV 1 and hPIV 2, hPIV 3 and hPIV 4, EV, hRV, and AdV. hPIV 3 likely comprises most of hPIV 3 + 4 incidences because hPIV 4 infections are detected infrequently and tend to be mild or asymptomatic[63].

We excluded hBoV, hPeV, MPn, and CPn from downstream analysis because probes for these pathogens were removed from our custom OpenArray panel in 2020. We also excluded ICV, EV.D68, measles, and mumps because these pathogens did not have a sufficient number of positive specimens to estimate daily incidence (< 200 positives from 2018 to 2022). A substantial number of specimens tested positive for SPn, a common commensal, with SPn detected in 26.7% of positive samples prior to March 2020 and 18.5% of positive samples after March 2020 (Figure. S2). We opted to not include SPn in the downstream analysis due to our inability to distinguish acute infections from chronic carriage.

All downstream data manipulation and analysis was performed using R version 4.3[64], unless otherwise noted.

### Syndromic surveillance data

We obtained respiratory syndromic surveillance data for King County, WA from the Rapid Health Information Network (RHINO) program at the Washington Department of Health (WA DOH) (Figure. S23), and for Washington state from the U.S. CDC Outpatient Influenza-like Illness Surveillance Network (ILINet)[65]. Syndrome criteria are described in the Supplementary Methods.

### Data on cell phone mobility, masking, and the stringency of non-pharmaceutical interventions

We obtained mobile device location data from SafeGraph (https://safegraph.com/), a data company that aggregates anonymized location data from 40 million devices, or approximately 10% of the United States population, to measure foot traffic to over 6 million physical places (points of interest) in the US. We estimated foot traffic to specific points of interest (POIs), movement within and between census block groups, and the in-flow of visitors residing outside of King County from November 2018 to June 2022, using SafeGraph's "Weekly Patterns" dataset, which provides weekly counts of the total number of unique devices visiting a POI from a particular home location. POIs are fixed locations, such as businesses or attractions. A "visit" indicates that a device entered the building or spatial perimeter designated as a POI. A "home location" of a device is defined as its common nighttime (18:00-7:00) census block group (CBG) for the past 6 consecutive weeks. We restricted our datasets to King County POIs that had been recorded in SafeGraph's dataset since January 2019. SafeGraph data were imported and processed using the SafeGraphR package[66].

To measure movement within and between CBGs ("neighborhoods") in King County, we extracted the home CBG of devices visiting points of interest (POIs) and limited the dataset to devices with home locations in the CBG of a given POI (within-neighborhood movement) or with home locations in CBGs outside of a given POI's CBG (between-neighborhood movement). To measure the inflow of visitors from other counties in Washington state or from out-of-state, we limited the dataset to devices visiting POIs in King County with home locations in other WA counties or in other US states, respectively. To measure foot traffic to specific categories of POIs, we aggregated daily visits to POIs by North American Industry Classification System (NAICS) category without considering the home locations of devices visiting these POIs. To adjust for variation in SafeGraph's device panel size over time, we divided Washington's census population size by the number of devices in SafeGraph's panel with home locations in Washington state each month and multiplied the number of daily or weekly visitors by that value. For each mobility indicator, we summed adjusted daily or weekly visits across POIs and measured the percent change in movement over time relative to the average movement observed in all of 2019, excluding national holidays.

Daily data on the percentage of devices staying home in King County were obtained from SafeGraph's Social Distancing Metrics and Meta Data for Good's Movement Range Maps. SafeGraph social distancing metrics were available from January 1, 2019, to April 16, 2021, and Meta Movement Range Maps were available from March 1, 2020, to May 22, 2022. Trends in the percentage of devices staying home were almost identical across the two data sources, though the percentage of devices staying home in the Meta dataset was lower than that observed in the SafeGraph dataset (Figure. S24). We added a scaling factor to the Meta indicator and joined the two-time series to create a single metric for our study period (Figure. S24). To reduce noise, we smoothed the joint time series with a centered 7-day moving average.

We obtained survey data on the daily percentage of King County residents wearing face masks in public from the Carnegie Mellon University Delphi Group COVIDcast API[27]. Masking data were collected as part of the COVID-19 Trends and Impact Survey conducted by the Delphi group in collaboration with Meta and a consortium of universities and public health officials[67]. The survey ran continuously from April 6, 2020, to June 25, 2022, with approximately 40,000 individuals in the United States participating every day. The survey included specific questions about masking from September 8, 2020, to June 25, 2022. We supplemented the COVIDcast King County masking data with COVIDNearYou survey data for Washington state[68] to extend the time series to June 2, 2020.

We extracted data collected by the Oxford COVID-19 Government Response Tracker (OxCGRT)[26] to measure variation in Washington state's government policies related to COVID-19 from March 1, 2020 to June 30, 2022. The OxCGRT database tracked publicly available information for policies related to closure and containment, health, and economic policy in 180 countries, recording policy responses on ordinal or continuous scales for 19 policy areas. We obtained daily values for the stringency index (Fig. 1), which combines all containment and closure indicators (C1-C8: school and university closures, workplace closures, cancellation of public events, restrictions on gatherings, closures of public transport, stay-at-home orders, restrictions on internal movement, and restrictions on international travel) and the H1 indicator (public information campaigns). The Oxford Stringency Index is based on policy mandates in place over time and does not measure the actual implementation of NPIs or population adherence to mandates[26].

### Reconstructing pathogen incidences

While SFS sampling is robust enough to provide granular (daily) surveillance data on the circulation of multiple pathogens, the diversity of SFS sampling schemes requires pre-processing to infer pathogen incidence. To properly reconstruct pathogen incidences through time, we considered the different populations sampled by SFS, particularly regarding age group, clinical setting, and the presence of respiratory symptoms (Figure. S25).

We first excluded samples with missing age or home address information (as reported by individuals participating in community surveillance or obtained through electronic hospital records), samples from individuals residing outside the greater Seattle region (King, Pierce, Snohomish, Kitsap, San Juan, Whatcom, Skagit, Island, Clallam, Jefferson, Mason, and Thurston counties), samples from individuals who were asymptomatic for respiratory illness, and samples from multiple testing of individuals. If an individual tested more than once in a 30-day period, we kept one result per pathogen in that period. If test results for all pathogens were consistent across the testing instances in the 30-day period, we kept the results from the first testing instance and discarded the subsequent instances. If an individual tested negative and then positive or tested positive then negative, we kept the result for the first positive testing instance and discarded the instances prior to or after that result. We also excluded samples collected as part of Public Health – Seattle & King County's (PHSKC) contact tracing efforts or through collaborations with community-based organizations.

Next, we used a three-step approach to control for sampling variation over time (Figure. S25). In the first step, we disaggregated daily pathogen presence and absence data derived from OpenArray testing by clinical setting (hospital or community) and age group (≥ 5 years or < 5 years). We then divided the number of positive samples for each pathogen by the total number of specimens tested in each setting and age stratum (Figure. S26). Daily proportion test-positive values for each age group $a$ and setting $s$ were then multiplied by the expected age distribution of cases for each pathogen in each setting, which were obtained from the U.S. Outpatient Influenza-like Illness Surveillance Network (ILINet)[65], the U.S. Influenza Hospitalization Surveillance Network (FluSurv-NET)[69], the Washington State Department of Health[70], or published literature (Table S6). For each pathogen, the daily adjusted proportion test-positive is calculated as:

$$\text{Adjusted Proportion Positive}_{a,s,t} = \frac{\text{Positive Tests}_{a,s,t}}{\text{Total Tests}_{a,s,t}} \times \mathbb{E}(\text{Prop.cases}_{a,s}),$$

$$(1)$$

where Positive Tests$_{a,s,t}$ and Total Tests$_{a,s,t}$ are the number of positive tests and total test volume for age group $a$ (≥ 5 years or < 5 years) in clinical setting $s$ (hospital or community) collected on day $t$, and

$\mathbb{E}(\text{Prop.cases}_{a,s})$ is the expected proportion of cases in age group $a$ in clinical setting $s$, based on external data sources.

In the second step, we combined pathogen proportion test-positive information from Equation (1) with citywide syndromic surveillance indicators for respiratory illnesses. Specifically, we multiplied the adjusted proportion test-positive data by a weekly indicator of the proportion of the King County population seeking care for respiratory illness at emergency departments (ED). Percent positive multiplied by the percentage of medical encounters attributable to respiratory illness is considered to be a more robust measure of respiratory virus activity than percent positive alone, and similar approaches have been used successfully to model influenza and seasonal coronavirus activity over multiple seasons (e.g.,[71,72]). We applied this adjustment separately to community and hospital data, wherein the daily adjusted proportion of test-positive values for each age group $a$ in clinical setting $s$ are multiplied by the weekly proportion of ED visits coded as general respiratory illness (all endemic viruses except influenza viruses), influenza-like illness, ILI (influenza viruses), or COVID-like illness, CLI (SARS-CoV-2) for age group $a$:

$$\text{Incidence}_{a,s,t} = \text{Adjusted Proportion Positive}_{a,s,t} \times \frac{\text{Respiratory illness visits}_{a,t}}{\text{Total visits}_{a,t}}$$

$$(2)$$

To produce an aggregate measure of daily incidence for each pathogen in the greater Seattle region, the third step entailed summing the age-specific incidences (< 5 years and ≥ 5 years) for each pathogen in each setting $s$ on day $t$ from Equation (2), rescaling age-combined hospital- and community-based incidences to fall between 0 and 1, and summing the separate community- and hospital-based incidences:

$$\text{Scaled Incidence}_{s,t} = \text{scale}(\text{Incidence}_{<5,s,t} + \text{Incidence}_{\geq 5,s,t}, 0, 1) \quad (3)$$

$$\text{Scaled Incidence}_t = \text{Scaled Incidence}_{hospital,t} + \text{Scaled Incidence}_{community,t}$$

$$(4)$$

We used this approach to estimate the daily incidences of 17 endemic pathogens from November 2018 to June 2022, including influenza A/H1N1, A/H3N2, and B viruses, RSV A and B, four seasonal coronaviruses, four human parainfluenza viruses, human metapneumovirus, rhinovirus, enterovirus, and adenovirus. We opted to estimate daily SARS-CoV-2 incidence from publicly available COVID-19 case data for King County[70] because SCAN did not test respiratory specimens for SARS-CoV-2 during May and June 2020 (Figure. S27).

### Statistical analysis

An overview of the statistical analyses and their various inputs is shown in Figure. S28.

**Transmission modeling.** For each pathogen, we estimated time-varying (instantaneous) reproduction numbers, $R_t$, by date of infection using the Epidemia R package[23,24]. $R_t$ can be expressed as the number of new infections $i$ on day $t$ relative to the cumulative sum of individuals infected $s$ days before day $t$, weighted by the current infectiousness of those individuals $g_k$[25,43]:

$$R_t = \frac{i_t}{\sum_{s<t} i_s g_{t-s}} \quad (5)$$

Epidemia implements semi-mechanistic Bayesian models using the probabilistic programming language Stan[73]. Instead of using a deterministic renewal process to propagate infections, we modelled new infections as unknown latent parameters, because the additional

variance around infections can account for uncertainty in initial growth rates, as well as superspreading events[23,24]. Transmission model specifications are described in the Supplementary Methods.

We evaluated changes in transmissibility during the two weeks before and after two major events in our study period: a major snowstorm in February 2019 and the initiation of COVID-19 social distancing measures in March 2020. For each pathogen, we used a nonparametric (two-sided) bootstrap test (1000 samples drawn with replacement) to estimate the ratio of mean $R_t$ values before and after each event and associated 95% confidence intervals (boot R package[74]).

**Cross-correlations between human behavior and pathogen transmission.** To measure dynamic associations between population behavior and pathogen transmissibility, we estimated rolling Spearman's rank cross correlations between mobility indicators and pathogen specific $R_t$ values. To avoid spurious correlations, we smoothed daily $R_t$ and mobility time series with centered 15-day moving averages. In all analyses, we weighted cross-correlations with an exponential decay such that observations at the edges of each time window were weighted approximately 50% less than observations at the window midpoint. This approach is reactive to recent changes in mobility and $R_t$ while still incorporating long-term trends. Due to limited testing at the beginning of the pandemic, we limited cross-correlations involving SARS-CoV-2 $R_t$ estimates to dates after February 28, 2020, when the first community-acquired case was confirmed, and cumulative confirmed cases exceeded 50. By March 1, 2020, the 95% credible intervals of SARS-CoV-2 $R_t$ were within 15% of the median value.

From Fall 2019 to Summer 2022, we computed cross-correlations between weekly pathogen specific $R_t$ values and the weekly percent change from baseline in mobility in rolling five-month windows. Although our original data have daily resolution, we chose to use weekly averages of $R_t$ and mobility due to the length of the timeframe analyzed and our desire to focus on broad long-term trends. We chose 5-month windows because this length of time provides a good trade-off between reducing noise and retaining a biologically relevant time window. In sensitivity analyses varying the length of rolling time window, shorter windows introduced more noise into the results (e.g., periods of alternating positive and negative correlations rather than consistent long-term trends) while longer windows diminished our ability to pinpoint when mobility most strongly correlated with $R_t$. During the pandemic period, we also estimated weekly cross-correlations and optimal lags between $R_t$ and the proportion of individuals masking in public (June 2020 to June 2022) and between $R_t$ and the Oxford Stringency Index (March 2020 to June 2022). During the 2018–2019 respiratory virus season, we estimated cross-correlations between daily $R_t$ and the daily percent change from baseline in mobility in rolling one-month windows, due to limited data at the start of that season (respiratory specimen collection began in November 2018) and to better capture the effects of the 12-day snowstorm in February 2019.

For each rolling window, we estimated weighted cross-correlations between mobility and $R_t$ at different lags (up to 4 weeks for 5-month rolling windows and up to 21 days for one-month rolling windows) and extracted the maximum (absolute) coefficient value and the lag (in weeks or days) at which this value occurred (the 'optimal lag'). Negative lag values indicate behavior leads $R_t$, and positive lag values indicate $R_t$ leads behavior. A lag of 0 indicates that two time series are in phase (i.e., synchronous). To generate monthly cross-correlations and lags, we averaged the correlation coefficients and optimal lags of window midpoints that fell within a given calendar month. As an example, for five-month rolling windows each month's statistics are an average of correlation coefficients and lags for dates falling 10 weeks prior to and 10 weeks following each week in that month.

To test the statistical significance of cross-correlations for each rolling window, we used a block bootstrap approach to generate 1000 samples of each mobility time series shuffled in two-week increments (tseries R package[75]) and recomputed cross-correlations between $R_t$ and mobility for each replicate, yielding a null distribution of 1000 cross-correlations. Cross-correlations between $R_t$ and mobility indicators were considered statistically significant when observed coefficients were outside the bounds of the null distribution's 90% interval. We performed bootstrapped cross-correlations using the high-performance computational resources of the Biowulf Linux cluster at the National Institutes of Health, with R version 4.2. Figures of monthly cross-correlations and optimal lags show both leading and lagging relationships between mobility and $R_t$, while figures of rolling cross-correlations are constrained to lags less than 1 to reduce noise and focus on synchronous or leading relationships.

As a sensitivity analysis, we estimated the daily transmissibility of the ancestral SARS-CoV-2 virus and each major variant of concern (VOC), using generation intervals, incubation periods, and reporting delays specific to each lineage, and computed rolling cross-correlations between VOC-specific $R_t$ values and behavioral indicators. Most VOC time series were too short to measure dynamic changes in correlations between $R_t$ and behavior, likely because VOC-specific analyses could not include the period immediately preceding increases in $R_t$.

**Multivariable generalized additive regression models.** For each pathogen, we used generalized additive models (GAMs) to measure non-linear relationships between mobility and $R_t$ and to assess the relative importance of different behavioral indicators in predicting $R_t$ during key epidemiological timepoints (see Supplementary Methods for specific dates). We used the mgcv R package[76] to fit each GAM with a Gamma error distribution and log link. Methodological details are described in the Supplementary Methods.

## Ethics oversight

The Seattle Flu Study and Greater Seattle Coronavirus Assessment Network were approved by the Institutional Review Board of the University of Washington (protocols #00006181 and #000010432). At the time of enrollment, participants provided informed consent for the respiratory sample and metadata collection and for the secondary use, banking, and/or future sharing of de-identified data for research purposes. These IRB protocols explicitly approve the use of the surveillance data for secondary research and do not impose restrictions on the specific types of secondary research that can be conducted or the external data sources that can be analyzed in tandem with the surveillance data. In accordance with UW IRB approval, informed consent for residual samples and clinical data collection was waived, as these samples were already collected as part of routine clinical care, and it was not possible to re-contact these individuals. IRB exemption for the use of non-publicly available aggregated respiratory syndromic surveillance data for King County, WA, was approved by the Washington State Institutional Review Board (Exempt Determination #2022-004). The human cellphone mobility data are aggregated and anonymous and were freely available to academic researchers prior to the start of this study; thus, these data do not constitute human subjects research. We did not collect cellphone data from surveillance study participants, and there is no individual-level linkage between the mobility and surveillance datasets. Individual-level linkage between these two datasets is not possible, given that the mobility data are aggregated and anonymous/de-identified. All other data sources pertaining to humans are aggregated, anonymous, and openly available. This study followed the Strengthening the Reporting of Observational Studies in Epidemiology (STROBE) reporting guidelines for cross-sectional studies.

**Reporting summary**

Further information on research design is available in the Nature Portfolio Reporting Summary linked to this article.

## Data availability

Aggregated epidemiological and mobility data that support the findings of this study can be accessed at https://doi.org/10.5281/zenodo.11044821[77] and https://github.com/aperofsky/seattle_mobility_rt. Access to deidentified individual-level study participant data requires a signed data access agreement with the Seattle Flu Alliance and can be made available to researchers whose proposed use of the data is approved by study investigators. Requests for data access should be submitted to data@seattleflu.org. Some mobility metrics were generated using SafeGraph Weekly Patterns and Social Distancing datasets, which were originally made freely available to academics in response to the COVID-19 pandemic. The SafeGraph Weekly Patterns dataset is currently available to academics for non-commercial use through an institutional university subscription or individual subscription to Dewey (https://www.deweydata.io/). The data access agreement with Dewey does not permit sharing of the raw data. Mobility data from Meta Data for Good Movement Range Maps are publicly accessible through the Humanitarian Data Exchange (https://data.humdata.org/dataset/movement-range-maps). SafeGraph social distancing data and Meta Data for Good survey data on masking are publicly accessible through the Carnegie Mellon Delphi group's COVIDcast Epidata API (https://cmu-delphi.github.io/delphi-epidata/api/covidcast.html). Data on the stringency of non-pharmaceutical interventions in US states are publicly accessible through the Oxford COVID-19 Government Response Tracker (https://github.com/OxCGRT/covid-policy-tracker). Aggregated influenza syndromic and virologic surveillance data for Washington state are publicly accessible through the US Centers for Disease Control and Prevention (CDC) FluView Interactive dashboard (https://www.cdc.gov/flu/weekly/fluviewinteractive.htm). Aggregated respiratory syndromic surveillance data for King County, WA are not publicly available and were provided by the Rapid Health Information Network (RHINO) program at the Washington Department of Health (WA DOH). Access for research purposes requires a signed data-sharing agreement with WA DOH and exemption approval from the Washington State Institutional Review Board. Requests for data access should be submitted to RHINO@doh.wa.gov. Data on COVID-19 cases in King County, WA are publicly accessible through the WA DOH COVID-19 dashboard (https://doh.wa.gov/emergencies/covid-19/data-dashboard). Data on COVID-19 vaccination in King County, WA are publicly accessible through the Public Health – Seattle & King County COVID-19 Vaccination dashboard (https://kingcounty.gov/en/dept/dph/health-safety/disease-illness/covid-19/data/vaccination). Nextstrain-curated SARS-CoV-2 sequence metadata can be downloaded via the Nextstrain CLI tool (https://docs.nextstrain.org/projects/cli/en/stable/). Daily records of precipitation, temperature, and humidity in Seattle, WA are publicly accessible through the National Centers for Environmental Information's U.S. Local Climatological Database (https://www.ncei.noaa.gov/products/land-based-station/local-climatological-data).

## Code availability

Code to reproduce the results and figures in this study is available at https://doi.org/10.5281/zenodo.11044821[77] and https://github.com/aperofsky/seattle_mobility_rt.

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

## Acknowledgements
We thank the entire Seattle Flu Study (SFS) and Greater Seattle Coronavirus Assessment Network (SCAN) team for their hard work and dedication to these projects and the study participants for their participation in this research. We also thank Public Health – Seattle & King County for their contributions to the SCAN study and for providing samples collected at King County COVID-19 drive-through testing sites, the Tacoma-Pierce County Health Department for funding the collection and testing of SCAN respiratory specimens in Pierce County, and the Rapid Health Information Network (RHINO) program at the Washington Department of Health for providing syndromic surveillance data. We thank Dr. Jeff Duchin for helpful comments on the manuscript and the Division of International Epidemiology and Population Studies (DIEPS) of the Fogarty International Center and the Bedford Lab at Fred Hutch for useful discussions. This study used the high-performance computational resources of the Biowulf Linux cluster at the US National Institutes of Health (http://biowulf.nih.gov). Funding for the Seattle Flu Study and Greater Seattle Coronavirus Assessment Network (SCAN) was provided by Gates Ventures and the Howard Hughes Medical Institute. SCAN samples collected in Pierce County were funded by the Tacoma-Pierce County Health Department. ACP, CLH, SB, RP, CM, DR, BC, KSF, KK, BP, ZA, EM, LRS, JSt, LG, PDH, AW, JSh, TB, HYC, and LMS received third-party support from Gates Ventures through the Brotman Baty Institute during the conduct of the study. ACP, LMS, and TB are supported by CDC contract 75D30122C14368. RB and MF are employees of the Institute for Disease Modeling, a research group within and solely funded by the Bill and Melinda Gates Foundation. JSh and TB are supported by the Howard Hughes Medical Institute. CV is supported by the in-house research division of the Fogarty International Center, US National Institutes of Health. For samples collected through mechanisms other than SCAN, the funders had no role in any aspect of the study. Gates Ventures participated in the design of SCAN by providing input on the study screener and eligibility criteria but had no role in the conduct of SCAN, the collection, management, analysis, or interpretation of SCAN data, the preparation, review, or approval of this manuscript, or the decision to submit the manuscript for publication. No other funders were involved in any aspect of SCAN. Disclaimer: The findings and conclusions in this report are those of the authors and do not necessarily represent the official position of the US National Institutes of Health or the US government.

## Author contributions
Conceptualization: A.C.P., M.F., J.Sh., T.B., H.Y.C., J.A.E., L.M.S., and C.V. Methodology: A.C.P, C.L.H., R.B., P.D.H., M.F., J.Sh., T.B., H.Y.C., J.A.E., L.M.S. and C.V. Software: A.C.P., C.L.H., R.B., C.M., D.R., B.C., M.T., K.S.F., K.K., B.P., J.L., T.R.S., J.St., A.A., M.F. and C.V. Validation: A.C.P., C.L.H., R.B., E.M., L.R.S., L.G., P.D.H. and L.M.S. Formal analysis: A.C.P. Investigation: A.C.P. Resources: A.A., M.L.J., J.Sh., T.B., H.Y.C., J.A.E., and L.M.S. Data acquisition or curation: A.C.P., C.L.H., R.B., C.M., D.R., B.C., M.T., K.S.F., K.K., B.P., Z.A., J.L., T.R.S., E.M., L.R.S., J.St., L.G., P.D.H., A.A., A.W., M.L.J., M.F., J.Sh., T.B., H.Y.C., J.A.E., L.M.S., and C.V. Writing—original draft: A.C.P. Writing—review & editing: A.C.P, C.L.H., R.B., S.B., R.P., M.F., H.Y.C., J.A.E., L.M.S., C.V. Visualization: A.C.P. Supervision: S.B., R.P., K.K., Z.A., J.St., P.D.H., M.F., J.Sh., T.B., H.Y.C., J.A.E., L.M.S. and C.V. Project administration: S.B., R.P., K.K., Z.A., J.St., P.D.H., M.F, J.Sh., T.B., H.Y.C., J.A.E., L.M.S. and C.V. Funding acquisition: A.C.P., C.L.H., S.B., R.P., A.W., J.Sh., T.B., H.Y.C., J.A.E., L.M.S. and C.V. All authors have seen and approved the manuscript.

## Competing interests
CLH received personal fees from Sanofi outside the submitted work. MLJ received funding as a contractor to Merck & Co. AW received clinical trial support to their institution from Pfizer, Ansun Biopharma, Allovir, and GlaxoSmithKline/Vir, personal fees from Vir and GlaxoSmithKline, and grants from Amazon outside the submitted work. JAE received grants from Pfizer, AstraZeneca, Merck, and GlaxoSmithKline and personal fees from Pfizer, AstraZeneca, GlaxoSmithKline, Merck, Meissa Vaccines, Moderna, and Sanofi Pasteur outside the submitted work. HYC received personal fees from Ellume, the Bill and Melinda Gates Foundation, Vindico, Abbvie, Merck, and Pfizer, research funding from Gates Ventures and Sanofi Pasteur, and support and reagents from Ellume and Cepheid outside the submitted work. CV received honoraria from Elsevier outside the submitted work. All other authors declare they have no competing interests.

## Additional information

[1]Brotman Baty Institute for Precision Medicine, University of Washington, Seattle, WA, USA. [2]Fogarty International Center, National Institutes of Health, Bethesda, MD, USA. [3]PandemiX Center, Department of Science & Environment, Roskilde University, Roskilde, Denmark. [4]Institute for Disease Modeling, Bill & Melinda Gates Foundation, Seattle, WA, USA. [5]Vaccine and Infectious Disease Division, Fred Hutchinson Cancer Center, Seattle, WA, USA. [6]Department of Genome Sciences, University of Washington, Seattle, WA, USA. [7]Seattle Children's Research Institute, Seattle, WA, USA. [8]Department of Pediatrics, University of Washington, Seattle, WA, USA. [9]EpiAssist LLC, Seattle, WA, USA. [10]Howard Hughes Medical Institute, Seattle, WA, USA. [11]Division of Allergy and Infectious Diseases, Department of Medicine, University of Washington, Seattle, WA, USA. ✉e-mail: acperof@uw.edu

