## [Peer Review File · Nature Communications]

Impacts of human mobility on the citywide transmission dynamics of 18 respiratory viruses in pre- and post-COVID-19 pandemic yearsREVIEWER COMMENTS

Reviewer #1 (Remarks to the Author):

The study investigates the association between human mobility (measured through mobile phone-derived metrics) and the transmission of 17 common respiratory viruses in Seattle, WA over a 4-year period, including the COVID-19 pandemic.

The authors identified epidemic phases when mobility indicators were associated with disease dynamics, and characterized what specific indicators were most (or least) predictive of transmission, for all the viruses under study.

My expertise is mainly related to the analysis of mobile phone-derived indicators and I will focus my comments on the aspects related to this part of the manuscript.

Overall, this is a solid work based on an extensive analysis of several mobile phone indicators, and the collection of a vast range of epidemiological surveillance data. This makes the paper an important and novel contribution to the field, despite many other studies explored the association between mobile phone-derived mobility indicators and epidemic dynamics. Previous studies have indeed generally considered a single pathogen at a time.

I believe the paper has enough merit to be published in Nature Communications. I have only a few comments that I consider minor, and I hope will help to improve the manuscript.

1. Colocation and mixing data. In the discussion the authors acknowledge that “spatial co-location data may better approximate the interpersonal contacts” but they don’t provide a reference for this and cite the CoMix survey as a potential alternative to solve this issue. First, it’s true that co-location data has been shown to be more informative in predicting the spread of SARS-CoV-2 than movement data. Two recent studies can support this claim:

- Delussu, F. et al. 2023. The limits of human mobility traces to predict the spread of COVID-19: A transfer entropy approach. PNAS nexus, 2(10), pgad302.
- Crawford FW, et al. 2022. Impact of close interpersonal contact on COVID-19 incidence: evidence from 1 year of mobile device data. Sci Adv. 8(1):5499.

Second, Meta’s colocation maps are publicly available and could be used to test this claim in the present study as well. Colocation data from Meta measure the probability that two users from two locations are found in the same location at the same time, on a 600x600 meters grid. However, they are provided on a weekly basis, so this would need some adjusting with respect to the current analysis.

2. Geographic scope. One limitation of the study is that its geographic scope is quite narrow, as the analysis focused on a single large metropolitan area of the US. The authors acknowledged such limitations, mainly considering the potential effects of within-city variations. At the same time, I would suggest acknowledging potential differences across the rural-urban divide, since it has been observed that mobility data may be less informative in sparsely populated areas (see Delussu et al. 2023) while the association is stronger in urban counties (see Kishore, N., et al 2022. Evaluating the reliability of mobility metrics from aggregated mobile phone data as proxies for SARS-CoV-2 transmission in the USA: a population-based study. The Lancet Digital Health, 4(1), e27-e36.)

3. Forecasting models. I commend the addition of the forecasting analysis which adds an important piece to the puzzle, still, I am not fully convinced of the interpretation. From Table S3, it seems that using mobility indicators in the forecasting models not only does not improve the accuracy with respect to the simple AR models, but could worsen it. My conclusion would be that the additional information provided by mobility indicators with respect to the past knowledge of disease incidence to forecast future disease trends is very limited and often not significant. This is fully in line with the results found in the study by Delussu et al. 2023 which were based on different metrics and a different approach.

Overall, even if correlation remains high (which however is not the metric commonly used to evaluate forecasts’ accuracy since it can easily take high values) I would conclude that when it

comes to forecasting future disease incidence the usefulness of mobility data is often limited.

4. GAM. To evaluate the association between mobility and disease, the authors used Generalized Additive Models. I would suggest adding a more detailed description of how a GAM is formulated, in mathematical terms and how is it able to capture non-linear relationships between two timeseries. I could not find this in the Methods but I think it is important to provide a full description of the approach for the readers.

Reviewer: Michele Tizzoni

Reviewer #2 (Remarks to the Author):

The impacts of human mobility and its changes on the transmission of SARS-CoV-2 and other respiratory infections such as seasonal influenza have been widely investigated during the COVID-19 pandemic. However, to what extent human mobility affected the dynamics of respiratory viruses before and after the pandemic remains less quantified. The research conducted by Perofsky A. et al aims to address this question by using a comprehensive and longitudinal dataset of respiratory pathogen surveillance/testing, mobile device geolocation-based human mobility in various settings, and public health interventions in Seattle from 2018 to 2022. Their findings further evidence the assumption that human mobility (or a proxy of contact rate) and the relevant travel interventions can reduce the spread and affect the rebound of both endemic and emerging respiratory viruses, with heterogeneity in the timing and lagged effects. Comments below could be addressed for improving this study.

Major comments:

1. The results show that in the early stage of the pandemic in 2020, mobility was a positive, leading indicator of transmission of all endemic viruses but negatively correlated with SARS-CoV-2 activity. However, this may be true given the underreporting or the very low number of cases at the early stage of the COVID-19 outbreak. This might be related to the margin effect of R_t estimation (ie. high initial R_t values) using a dynamic model.
2. Figs 4 and 5 (also other similar Figs in SI) seem quite complicated and hard to interpret, by including the positive lags between mobility and R_t . Theoretically, transmission can be mitigated by reducing travel and contact rates, either due to the governmental health policy or the concern of transmission in the public. Moreover, this paper aims to explore the impact of mobility on transmission. I think authors could just focus on the leading impact of human behaviour on R_t , ie. negative lags between behaviours and infection date-based R_t , and simplify figures and results.
3. Are the data distributions suitable for Pearson cross-correlations and t-tests used in this study? Maybe use a non-parametric test, e.g Mann-Whitney U test? Instead of using GAMs, the distributed lag non-linear models (DLNM) framework can be used to simultaneously explore and represent non-linear exposure-response dependencies and delayed effects.
4. A diagram will help understand the process of assembling, testing and adjusting data and reconstructing incidence. Another diagram will be also used to illustrate the statistical analysis and modelling framework of estimating R_t , exploring correlations, and GAMs.
5. Even though mobility might be the same, the co-circulation and the different transmissibility of SARS-CoV-2 VOCs as well as the immunity could modify the effects of mobility and interventions. To partially address this issue, the model can include the proportion of VOCs and the vaccination rate or the cumulated proportion of infections among the population.
6. Climatic factors, the interactions and competition between different viruses can also affect the transmission patterns. A multiple-year model including these factors across non-epidemic and epidemic seasons or waves should be considered.
7. This study makes extensive use of incidence reconstruction, weekly aggregation, rolling and different time windows. It is unclear whether these data manipulations alter intrinsic correlations. I wonder what the overall correlation and modelling results using sample positivity would look like without all of these processes.

Minor comments:

8. The manuscript title may be revised to better reflect that this study covers 17 pathogens, the 4-year period of pre-, during and post- pandemic years, the investigation of other human behaviours, as well as the study location.

9. Table 1 - 'Mean age': What do the numbers in brackets represent?

10. Please provide formulas about how to reconstruct daily incidence for each pathogen and adjust it for testing volume, demography, and syndromic surveillance rates across clinical settings.

11. Authors limited the analysis to 17 viruses with ≥ 400 positive samples each during 2018-2022. This sounds a bit arbitrary. Maybe also test another number e.g 300 or 500 positive samples?

12. Table S1 only mentions the number of pathogens that were tested from May 5, 2019. How many pathogens were targeted and detected between Nov 19, 2018 – May 4, 2019?

13. SafeGraph and Meta Data for Good data have different data collection methods and use different baselines. How consistent are they? Maybe provide a diagram to illustrate this.

14. The overall OxCGRT NPI stringency index data also include mobility-related and face masking measures. This study can use OxCGRT's measure-specific index to avoid the multicollinearity in the model.

Response to Reviewers

Perofsky et al., “Impacts of human mobility on the citywide transmission dynamics of 18 respiratory viruses in pre- and post-COVID-19 pandemic years” (NCOMMS-23-60234)

REVIEWER COMMENTS

Reviewer #1 (Remarks to the Author):

The study investigates the association between human mobility (measured through mobile phone-derived metrics) and the transmission of 17 common respiratory viruses in Seattle, WA over a 4-year period, including the COVID-19 pandemic.

The authors identified epidemic phases when mobility indicators were associated with disease dynamics, and characterized what specific indicators were most (or least) predictive of transmission, for all the viruses under study.

My expertise is mainly related to the analysis of mobile phone-derived indicators, and I will focus my comments on the aspects related to this part of the manuscript.

Overall, this is a solid work based on an extensive analysis of several mobile phone indicators, and the collection of a vast range of epidemiological surveillance data. This makes the paper an important and novel contribution to the field, despite many other studies explored the association between mobile phone-derived mobility indicators and epidemic dynamics. Previous studies have indeed generally considered a single pathogen at a time.

I believe the paper has enough merit to be published in Nature Communications. I have only a few comments that I consider minor, and I hope will help to improve the manuscript.

1. Colocation and mixing data. In the discussion the authors acknowledge that “spatial co-location data may better approximate the interpersonal contacts” but they don’t provide a reference for this and cite the CoMix survey as a potential alternative to solve this issue. First, it’s true that co-location data has been shown to be more informative in predicting the spread of SARS-CoV-2 than movement data. Two recent studies can support this claim:

- Delussu, F. et al. 2023. The limits of human mobility traces to predict the spread of COVID-19: A transfer entropy approach. PNAS nexus, 2(10), pgad302.**
- Crawford FW, et al. 2022. Impact of close interpersonal contact on COVID-19 incidence: evidence from 1 year of mobile device data. Sci Adv. 8(1):5499.**

Second, Meta’s colocation maps are publicly available and could be used to test this claim in the present study as well. Colocation data from Meta measure the probability that two users from two locations are found in the same location at the same time, on a 600x600 meters grid. However, they are provided on a weekly basis, so this would need some adjusting with respect to the current analysis.

Response: We thank the reviewer for pointing us to Delussu et al, 2023 and Crawford et al. 2022, which support our comment in the discussion that spatial colocation data may better approximate interpersonal contacts that are relevant for disease transmission. We have added these references to the paragraph about the limitations of mobile device location data (Lines 393-395).

We agree that it would be ideal to test if spatial colocation correlates more strongly with Rt than large-scale population movements inferred from foot traffic patterns. After some difficulty finding a public dataset for the Meta Colocation Maps, and communication with the Reviewer via the Editor, we learnt that this dataset is not publicly available but can be obtained through the Meta Data for Good Partner Portal, after requesting access and signing a DUA. The Colocation Map data for our study period are discontinued and no longer available in the portal, due to fundamental changes in how the colocation metric is calculated. We contacted the manager of research datasets for Meta Data for Good, and the data science team attempted to recover the discontinued dataset for us. Their approach was not successful so they recommended contacting other researchers from our institution who have used the Colocation Map data to ask if they'd be able to share the old dataset (per Meta's DUA, researchers from different institutions cannot share data). Unfortunately, the other UW researchers did not have the discontinued Colocation dataset. Although we would have liked to incorporate a spatial colocation metric into our analysis, we could not obtain the discontinued dataset recommended by the reviewer and do not know of other spatial colocation datasets that are publicly accessible.

In the paragraph on limitations of cellphone mobility data, we have added (Lines 395-397):

“Although we would have liked to incorporate a spatial colocation metric into our analysis, at the time of writing, Meta Data for Good’s Colocation Map dataset is discontinued for our study period, and we did not know of other spatial location datasets that are publicly accessible.”

2. Geographic scope. One limitation of the study is that its geographic scope is quite narrow, as the analysis focused on a single large metropolitan area of the US. The authors acknowledged such limitations, mainly considering the potential effects of within-city variations. At the same time, I would suggest acknowledging potential differences across the rural-urban divide, since it has been observed that mobility data may be less informative in sparsely populated areas (see Delussu et al. 2023) while the association is stronger in urban counties (see Kishore, N., et al 2022. Evaluating the reliability of mobility metrics from aggregated mobile phone data as proxies for SARS-CoV-2 transmission in the USA: a population-based study. The Lancet Digital Health, 4(1), e27-e36.)

Response: Thank you for pointing out this limitation of our study. We now mention geographic scope in the paragraph about the limitations of cell phone mobility data, referencing Delussu et al. 2023, Kishore et al. 2022, and Jewell et al. 2021 (Lines 390-393).

“Second, relationships between cellphone mobility and transmission are weaker in more sparsely populated areas¹⁻³, due to differences in the data generation process and representativeness between urban and rural locations. Because our study is limited to a single metropolitan region, our findings may not be applicable to rural locations.”

3. Forecasting models. I commend the addition of the forecasting analysis which adds an important piece to the puzzle, still, I am not fully convinced of the interpretation. From Table S3, it seems that using mobility indicators in the forecasting models not only does not improve the accuracy with respect to the simple AR models but could worsen it. My conclusion would be that the additional information provided by mobility indicators with respect to the past knowledge of disease incidence to forecast future disease trends is very limited and often not significant. This is fully in line with the results found in the study by Delussu et al. 2023 which were based on different metrics and a different approach.

Overall, even if correlation remains high (which however is not the metric commonly used to evaluate forecasts' accuracy since it can easily take high values) I would conclude that when it comes to forecasting future disease incidence the usefulness of mobility data is often limited.

Response: Thank you for pointing this out. We have updated our analysis to consider models with autoregressive (AR) terms as “baseline” models to which other models with various combinations of AR, mobility, climatic, and virus-virus interaction terms are compared. Our results now focus on how mobility can improve prediction accuracy, and we tone down statements concerning the utility of mobility data in forecasting future transmission dynamics. We also cite that our results are consistent with Delussu et al. 2023.

For SARS-CoV-2 Rt, we found that models with mobility and AR terms had greater prediction accuracy compared to baseline models during the stay-at-home period in 2020 and during 2021-2022 (the time period encompassing COVID-19 vaccination and variant emergence), but not for the whole study period combined. For hRV and AdV, models with mobility and AR terms did not improve prediction accuracy over baseline models during any time period.

In the main text, Lines 259-263: *“When assessing model accuracy over the entire study period, models including mobility, climatic variables, or viral interference did not outperform baseline models for any of the three viruses (Figs. S19-S21; Table S3). Thus, although mobility data can provide small to moderate benefits to prediction accuracy, the additional information provided by past population movements is limited in comparison to knowledge of past disease incidence³.”*

In the supplementary results, third paragraph: *“Overall, we found that prior disease activity alone is most beneficial for accurately projecting future transmission dynamics. Tracking mobility behavior is not essential for forecasting respiratory virus transmission, and the inclusion of mobility data can even be detrimental to prediction accuracy, depending on the pathogen and time period (Tables S3-S5). Monitoring major changes in mobility could still be helpful for general situational awareness and planning purposes in the early stages of an emerging disease outbreak, when testing capacity is low and the true incidence of the disease is unknown. However, prior information on mobility trends is unlikely to provide a net benefit to prediction accuracy when an epidemic is widely established in a population. This finding is consistent with another study that used a different modeling approach and set of mobility metrics to forecast COVID-19 cases and deaths in Europe³.”*

We originally included Pearson’s correlation coefficients because this accuracy metric was used in the original study⁴ that published the forecasting methodology we use in our study. We agree that Pearson correlations are not a standard accuracy metric in forecasting and are inappropriate to use in this context, due to their tendency to take high values and the test’s assumption of linearity. We now focus on root-mean-squared-error (RMSE) and mean absolute error (MAE) in our results.

4. GAM. To evaluate the association between mobility and disease, the authors used Generalized Additive Models. I would suggest adding a more detailed description of how a GAM is formulated, in mathematical terms and how it is able to capture non-linear relationships between two time series. I could not find this in the Methods, but I think it is important to provide a full description of the approach for the readers.

Response: Our revised manuscript provides more details about our GAM approach in the supplementary methods (section “Multivariable generalized additive regression models”), including additional background information concerning how GAMs are able to capture nonlinear relationships and the formal mathematical equation for the best fit minimal models of Rt.

Reviewer #2 (Remarks to the Author):

The impacts of human mobility and its changes on the transmission of SARS-CoV-2 and other

respiratory infections such as seasonal influenza have been widely investigated during the COVID-19 pandemic. However, to what extent human mobility affected the dynamics of respiratory viruses before and after the pandemic remains less quantified. The research conducted by Perofsky A. et al aims to address this question by using a comprehensive and longitudinal dataset of respiratory pathogen surveillance/testing, mobile device geolocation-based human mobility in various settings, and public health interventions in Seattle from 2018 to 2022. Their findings further evidence the assumption that human mobility (or a proxy of contact rate) and the relevant travel interventions can reduce the spread and affect the rebound of both endemic and emerging respiratory viruses, with heterogeneity in the timing and lagged effects. Comments below could be addressed for improving this study.

Major comments:

1. The results show that in the early stage of the pandemic in 2020, mobility was a positive, leading indicator of transmission of all endemic viruses but negatively correlated with SARS-CoV-2 activity. However, this may be true given the underreporting or the very low number of cases at the early stage of the COVID-19 outbreak. This might be related to the margin effect of R_t estimation (ie. high initial R_t values) using a dynamic model.

Response: Thank you for pointing out this caveat concerning our finding that SARS-CoV-2 R_t lagged mobility during the early months of 2020. In response to this comment, we have both highlighted the potential issue in the discussion and revised our analyses to remove very early pandemic observations. Our revised manuscript now mentions that R_t estimates at the beginning of the pandemic may be upwardly biased, due to low case counts and limited testing capacity. Further, to minimize the potential influence of uncertain R_t estimates on cross-correlation coefficients, our revised manuscript limits SARS-CoV-2 R_t estimates to dates after February 28, 2020, when the first community acquired case was detected, and cumulative confirmed cases exceeded 50. By March 1, 2020, the 95% prediction intervals of R_t are within 15% of the median value.

Discussion, Lines 334-344: *“Mobility had a negative, lagging relationship with SARS-CoV-2 R_t during the early months of 2020, suggesting that Seattle residents adjusted their behavior in response to COVID-19 case counts or restrictions rather than the reverse. This result is consistent with a comprehensive analysis of US counties, which found that not all major cities (e.g., San Francisco) experienced strong positive associations between mobility and infection growth rates during the first wave². A caveat is that R_t estimates during this time were likely upwardly biased, due to low case counts, sampling biases, and rapid increases in testing capacity, and may not have fully captured the steepness of transmission declines after NPIs were implemented. To mitigate this issue, we limited SARS-CoV-2 R_t estimates to dates after February 28, 2020, when the first community-acquired case was detected, and cumulative confirmed cases exceeded 50. We also applied centered smoothing windows to case counts, R_t estimates, and mobility indicators so that cross-correlations captured broad signals that were accurately oriented in time^{5,6}.”*

Methods, Lines 584-587: *“Due to limited testing at the beginning of the pandemic, in cross-correlations we limited SARS-CoV-2 R_t estimates to dates after February 28, 2020, when the first community-acquired case was confirmed, and cumulative confirmed cases exceeded 50. By March 1, 2020, the 95% prediction intervals of SARS-CoV-2 R_t were within 15% of the median value.”*

2. Figs 4 and 5 (also other similar Figs in SI) seem quite complicated and hard to interpret, by including the positive lags between mobility and R_t . Theoretically, transmission can be mitigated by reducing travel and contact rates, either due to the governmental health policy or the concern of transmission in the public. Moreover, this paper aims to explore the impact of mobility on

transmission. I think authors could just focus on the leading impact of human behaviour on Rt, i.e. negative lags between behaviours and infection date-based Rt and simplify figures and results.

Response: We appreciate the reviewer's suggestions for ways to simplify our figures so that they are easier to interpret. For figures showing average monthly cross-correlations and optimal lags (e.g., Figure 4), we opted to keep both leading and lagging relationships between mobility and Rt, because showing only leading relationships (i.e., mobility leads Rt) removes our ability to visualize a key finding that mobility initially lagged SARS-CoV-2 Rt at the beginning of the pandemic. To simplify Figure 4, we limit the mobility metrics to those that have the strongest correlations with endemic pathogen Rt during Fall 2019; the original version of the figure showing all mobility metrics is now in the supplement (Figure S10). For figures showing rolling cross-correlations (e.g., Figure 5), we now constrain cross-correlations to leading or synchronous relationships between mobility and Rt, which reduces noise and improves interpretability for readers.

3. Are the data distributions suitable for Pearson cross-correlations and t-tests used in this study? Maybe use a non-parametric test, e.g. Mann–Whitney U test? Instead of using GAMs, the distributed lag non-linear models (DLNM) framework can be used to simultaneously explore and represent non-linear exposure–response dependencies and delayed effects.

Response: Thank you for pointing out that our data may violate the assumptions of some of the statistical tests we originally included in the paper.

For analyses measuring mean changes in Rt before and after two major events, the snowstorm in February 2019 and the State of Emergency declaration in late February 2020, the datasets are too small to test if the distributions of Rt values meet the assumption of normality for t-tests. In the revised manuscript, we now use nonparametric bootstrap tests of the ratio of two means (with 1000 samples) instead of t-tests of the ratio of two means (Lines 574-577).

Pearson correlation tests do not require normality to estimate the correlation coefficient r itself; however, the test requires that the joint distribution of X and Y is bivariate normal to make inferences about the relationship (e.g., to test the null hypothesis, $r = 0$). We do not think it is necessarily inappropriate to test for linear relationships between mobility and Rt, because our P-values for statistical significance are estimated via a nonparametric time series block bootstrap approach that shuffles the mobility time series 1000 times and assesses if observed correlations fall within the null distribution of correlation coefficients. In our revised manuscript we replace Pearson correlation tests with nonparametric Spearman rank correlation tests, which measure monotonic relationships between variables and do not assume linearity or an underlying distribution of the data. Although we found qualitatively similar results for Pearson and Spearman tests, we appreciate the reviewer's suggestion to examine the assumptions of our statistical tests more closely.

Concerning the reviewer's suggestion to consider DLNMs instead of GAMs, we found that the DLNM framework can be incorporated into GAMs⁷, by including a cross-basis function as a model covariate. DLNMs on their own are more restrictive than GAMs, because they require the user to select the parametric form of the functions expressing the dose-lag-response relationship⁷. We modified our GAMs to include the cross-basis of mobility and time lags of up to 14 days. We also tested tensor product smooths of mobility and time lags, which are similar to cross-basis functions⁷, and more well documented in R (via the “mgcv” package). Including the bivariate cross-basis or tensor product smooth of mobility and time lags reveals which lags of mobility have the strongest positive or negative relationships with Rt. However, including interactions between mobility and time lags removes our ability to visualize whether relationships between mobility and Rt are linear or nonlinear. As an example, see the figure below

showing the partial effects of tensor product smooths of mobility and time lags on RSV B Rt during the 2019-2020 season.

We also experienced issues with model convergence and overfitting when incorporating the cross-basis or tensor product smooth of mobility and time lags. Based on the DLNM literature, it seems that this approach is traditionally used in epidemiological studies assessing the cumulative lagged effects of pollutants on disease outcomes over the course of decades (e.g., Neophytou et al. 2018⁸). GAMs are already parameter heavy models, and our time series of Rt and mobility focus on the exponential growth phase of each epidemic wave (2- to 6-month time periods, depending on the pathogen and time period). Our issues with non-identifiability likely stem from having more covariates in the model than our dataset can realistically support. Thus, we opted to keep our simpler original analysis, which was intended to select the most important behavioral predictors of Rt and to visualize the shape and direction of relationships between mobility and Rt. As a sensitivity analysis we tested GAMs with lagged mobility covariates (e.g., 7-day or 14-day lags), but lagged mobility trends did not improve model fit over the original models that focused on synchronous relationships between mobility and Rt. However, in our forecasting models of pathogen Rt, we now include lagged mobility covariates.

4. A diagram will help understand the process of assembling, testing and adjusting data and reconstructing incidence. Another diagram will be also used to illustrate the statistical analysis and modelling framework of estimating Rt, exploring correlations, and GAMs.

Response: To help readers understand the process of reconstructing incidences, our revised manuscript includes mathematical formulas in the methods section (“Reconstructing pathogen incidences,” Lines 510 - 560) and a flowchart in the supplement (Figure S25). We have also added a diagram showing an overview of statistical analyses performed in our study and their various inputs (Figure S28).

5. Even though mobility might be the same, the co-circulation and the different transmissibility of SARS-CoV-2 VOCs as well as the immunity could modify the effects of mobility and interventions. To partially address this issue, the model can include the proportion of VOCs and the vaccination rate or the cumulated proportion of infections among the population.

Response: In our study, GAMs measuring the effects of mobility on SARS-CoV-2 Rt are separated by waves. Given the wave-by-wave analysis, the short time series of each wave, and collinearity/concurvity among covariates, it would be difficult to disentangle the individual effects of variant emergence, vaccination, and mobility on Rt with our current approach. Further, testing the effects of vaccination or variant emergence on Rt on a wave-by-wave basis could be “fraught with difficulty⁹,” given that pre-existing levels of natural immunity vary across individual waves. Lastly, Rt is estimated from the sum of prior infections weighted by the generation interval, with past infections propagating new infections. The weighting of cumulative incidence by the generation interval makes it so that infections that occurred

prior to the past few weeks (i.e., the length of SARS-CoV-2's serial interval) have very low or no weight in the estimation of present-day R_t . However, because cumulative incidence is intrinsically linked to the estimation of R_t , it could be considered a form of "circular" analysis¹⁰ to include it as a predictor of R_t in regression models.

We agree with the reviewer that exploring the effect of prior immunity would be important over the course of an entire epidemic wave. Although it is possible to adjust R_t for susceptible depletion with our semi-mechanistic modeling approach, the Epidemia framework cannot incorporate the waning of immunity or cross-immunity between variants. Thus, we opted to not explicitly account for changes in immunity in the R_t estimation process, because we believe it requires a fully mechanistic approach to be done properly. Further, our GAM analyses are restricted to timeframes spanning the exponential growth phase of each outbreak, when R_t exceeds 1 and susceptible depletion is limited, which reduces the confounding effects of depletion of susceptibles on R_t .

Our forecasting approach for SARS-CoV-2 R_t is not hindered by short time series and can better handle collinearity between covariates. In the revised manuscript we now incorporate covariates for cumulative vaccination coverage and variant circulation in models forecasting SARS-CoV-2 R_t .

6. Climatic factors, the interactions and competition between different viruses can also affect the transmission patterns. A multiple-year model including these factors across non-epidemic and epidemic seasons or waves should be considered.

Response: We have implemented the reviewer's suggestion for our forecasting analyses, with limited results (detailed below). We have refrained from exploring the effects of climatic factors and viral interference across all of our analyses, because it would require a much longer time series of incidence (several years of pre-pandemic data) or complex multi-pathogen mechanistic models accounting for viral seeding, the influx of susceptibles each season, and the emergence of antigenic variants (for SARS-CoV-2 and influenza viruses). Kissler et al. 2020¹¹ used a simple statistical model to estimate the effects of seasonality (modeled via splines) and cross-immunity on the circulation of two seasonal coronaviruses prior to the COVID-19 pandemic. However, their multi-year model focused on the epidemic period of each virus (i.e., did not include intervening periods with low or no circulation) and included 5 years of pre-pandemic data. Our study consists of 2 pre-pandemic seasons and a 2-year period of low circulation for most endemic viruses. Thus, it would be difficult to measure the effects of pathogen interactions or meteorological factors on transmission, given our limited dataset. Further, viral interference at the individual host level is well-established, but the epidemiological impact of this phenomenon is not well understood, even for the most well-studied pathogens in our dataset: influenza and RSV.

For our forecasting models of R_t , we focused on 3 pathogens with continuous circulation throughout our study period: rhinovirus, adenovirus, and SARS-CoV-2. These models include covariates for past viral activity, cellphone mobility, viral interference (the impact of SARS-CoV-2 circulation on rhinovirus or adenovirus R_t , and vice versa), and local environmental conditions (temperature, precipitation, and humidity). We found that climatic variables slightly improved predictions for SARS-CoV-2 R_t and adenovirus R_t during Seattle's stay-at-home period in 2020 but did not improve prediction accuracy for any of the three viruses over the course the entire study period. Covariates for viral interactions did not improve model performance for any of the three viruses.

7. This study makes extensive use of incidence reconstruction, weekly aggregation, rolling and different time windows. It is unclear whether these data manipulations alter intrinsic correlations. I wonder what the overall correlation and modelling results using sample positivity would look like without all of these processes.

Response: In the revised manuscript we provide more detailed justifications for our methodology, which is intended to capture intrinsic correlations and reduce observational noise in the data.

To estimate pathogen incidence, we chose to multiply the percentage of respiratory samples testing positive for a particular pathogen by the percentage of respiratory-like illness visits because this measure is considered to be a more robust estimate of respiratory virus activity than percent positive alone and has been used successfully in many epidemiological modeling studies¹¹⁻¹⁶. We smoothed incidences prior to R_t estimation so that changes in R_t are more representative of true rises or declines in transmission and less influenced by imperfect observation and reporting irregularities^{5,6,17}. In sensitivity analyses varying the degree of smoothing, estimating R_t from raw incidences caused model convergence issues and highly variable R_t estimates, consistent with the findings of Huisman et al. 2022⁶. Huisman et al. 2022 found that smoothing incidences prior to deconvolution causes slight misestimations of R_t during steep changes; however, the authors concluded that the benefit of improved model performance and more stable R_t estimates outweighs this issue. In the revised manuscript we now include a supplementary figure of unadjusted percent positive values to show readers how noisy the raw pathogen presence/absence data are (Figure S26).

Our analysis of rolling cross-correlations is intended to capture dynamic, biologically relevant trends in which mobility indicators are most strongly correlated with R_t , across different pathogens and time periods. For cross-correlations spanning Fall 2019 to June 2022, we chose to use weekly averages of R_t and mobility, rather than daily values, to reduce noise and focus on broad long-term trends. Additionally, it is difficult to visualize daily rolling cross-correlations over long time periods because it involves squishing many data points together. We chose 5-month rolling windows because this length of time provides a good trade-off between reducing noise and retaining a biologically relevant time window. In sensitivity analyses varying the length of rolling window, we found that shorter time windows introduced more noise into the results (e.g., alternating positive and negative correlations rather than consistent long-term trends) while longer windows diminished our ability to pinpoint when mobility most strongly correlated with R_t . The cross-correlation analysis for the 2018-2019 season uses daily estimates of R_t and mobility because the time window of analysis is short, and we wanted to capture the impact of a short interruption in movement (a major snowstorm) on R_t .

For consistency, the same smoothing window is applied to the time series of mobility and R_t for each pathogen and mobility indicator. If the observed correlations are spurious, we would not observe consistent patterns across pathogens during key epidemiological time points, such as the stay-at-home period in March 2020, the major snowstorm in February 2019, the first months of enveloped virus rebound in the spring and summer of 2021, and the Omicron BA.1 wave in late 2021.

Minor comments:

8. The manuscript title may be revised to better reflect that this study covers 17 pathogens, the 4-year period of pre-, during and post- pandemic years, the investigation of other human behaviours, as well as the study location.

Response: We appreciate this suggestion and would like to include more detail in our title; however, the journal limits the title to 15 words. We've added the number of pathogens to the title, if the Editor will allow us to go a little over the word limit.

9. Table 1 - 'Mean age': What do the numbers in brackets represent?

Response: Thank you for catching this. We now specify that the brackets enclose the standard deviation of age.

10. Please provide formulas about how to reconstruct daily incidence for each pathogen and adjust it for testing volume, demography, and syndromic surveillance rates across clinical settings.

Response: Our revised manuscript now includes formulas for each step of incidence reconstruction (Lines 510 - 560).

11. Authors limited the analysis to 17 viruses with ≥ 400 positive samples each during 2018-2022. This sounds a bit arbitrary. Maybe also test another number e.g. 300 or 500 positive samples?

Response: We agree that our original statement concerning limiting the analysis to pathogens with sufficient sampling did not provide enough information concerning the decision process for which pathogens we included. If we increase the threshold to 500 positives, hPIV 1 + 2 would no longer be included in the analysis. Decreasing the threshold to 300 positives would not add any pathogens to the analysis. Our custom OpenArray platform did not test for all pathogens for the entire duration of the study, which causes some pathogens to have low numbers of positives. We now provide more information in the methods concerning why certain pathogens were excluded (Lines 450-456).

In our revised manuscript, we now include non-rhinovirus enterovirus (EV), which we originally excluded because our laboratory assay cannot differentiate between single enterovirus infections and enterovirus-rhinovirus coinfections. Given that we do not exclude coinfections for the other pathogens, EV is now incorporated into our analysis.

12. Table S1 only mentions the number of pathogens that were tested from May 5, 2019. How many pathogens were targeted and detected between Nov 19, 2018 – May 4, 2019?

Response: Thank you for catching this. In the original manuscript we failed to include the first version of the OpenArray panel that spanned March to April 2019, which is now added to the Table S1. We clarified with our co-authors that testing did not begin until March 2019, even though sample collection started in November 2018. This information is now included in the caption of Table S1.

13. SafeGraph and Meta Data for Good data have different data collection methods and use different baselines. How consistent are they? Maybe provide a diagram to illustrate this.

Response: Our revised manuscript includes a figure showing how we combined these two data sources into one metric (Figure S24).

14. The overall OxCGRT NPI stringency index data also include mobility-related and face masking measures. This study can use OxCGRT's measure-specific index to avoid the multicollinearity in the model.

Response: Although the Oxford Stringency Index (OSI) includes policy mandates that could affect mobility it does not measure the actual implementation of policy measures or adherence to mandates. The particular index we use does not include an indicator for masking. We have added more information about the specific policies included in the OSI to the methods (Lines 504-509).

“We obtained daily values for the stringency index (Fig. 1), which combines all containment and closure indicators (C1-C8: school and university closures, workplace closures, cancellation of public events, restrictions on gatherings, closures of public transport, stay-at-home orders, restrictions on internal

movement, and restrictions on international travel) and the HI indicator (public information campaigns). The Oxford Stringency Index is based on policy mandates in place over time and does not measure the actual implementation of NPIs or population adherence to mandates¹⁸.”

References

- 1 Kishore, N. *et al.* Evaluating the reliability of mobility metrics from aggregated mobile phone data as proxies for SARS-CoV-2 transmission in the USA: a population-based study. *Lancet Digit Health* **4**, e27-e36 (2022). [https://doi.org/10.1016/S2589-7500\(21\)00214-4](https://doi.org/10.1016/S2589-7500(21)00214-4)
- 2 Jewell, S. *et al.* It's complicated: characterizing the time-varying relationship between cell phone mobility and COVID-19 spread in the US. *NPJ Digit Med* **4**, 152 (2021). <https://doi.org/10.1038/s41746-021-00523-3>
- 3 Delussu, F., Tizzoni, M. & Gauvin, L. The limits of human mobility traces to predict the spread of COVID-19: A transfer entropy approach. *PNAS Nexus* **2**, pgad302 (2023). <https://doi.org/10.1093/pnasnexus/pgad302>
- 4 Yang, S., Santillana, M. & Kou, S. C. Accurate estimation of influenza epidemics using Google search data via ARGO. *Proc Natl Acad Sci U S A* **112**, 14473-14478 (2015). <https://doi.org/10.1073/pnas.1515373112>
- 5 Gostic, K. M. *et al.* Practical considerations for measuring the effective reproductive number, Rt. *PLoS Comput Biol* **16**, e1008409 (2020). <https://doi.org/10.1371/journal.pcbi.1008409>
- 6 Huisman, J. S. *et al.* Estimation and worldwide monitoring of the effective reproductive number of SARS-CoV-2. *Elife* **11** (2022). <https://doi.org/10.7554/eLife.71345>
- 7 Gasparrini, A., Scheipl, F., Armstrong, B. & Kenward, M. G. A penalized framework for distributed lag non-linear models. *Biometrics* **73**, 938-948 (2017). <https://doi.org/10.1111/biom.12645>
- 8 Neophytou, A. M. *et al.* Exposure-Lag-Response in Longitudinal Studies: Application of Distributed-Lag Nonlinear Models in an Occupational Cohort. *Am J Epidemiol* **187**, 1539-1548 (2018). <https://doi.org/10.1093/aje/kwy019>
- 9 Jewell, N. P. & Lewnard, J. A. On the use of the reproduction number for SARS-CoV-2: Estimation, misinterpretations and relationships with other ecological measures. *J R Stat Soc Ser A Stat Soc* (2022). <https://doi.org/10.1111/rssa.12860>
- 10 Makin, T. R. & Orban de Xivry, J. J. Ten common statistical mistakes to watch out for when writing or reviewing a manuscript. *Elife* **8** (2019). <https://doi.org/10.7554/eLife.48175>
- 11 Kissler, S. M., Tedijanto, C., Goldstein, E., Grad, Y. H. & Lipsitch, M. Projecting the transmission dynamics of SARS-CoV-2 through the postpandemic period. *Science* **368**, 860-868 (2020). <https://doi.org/10.1126/science.abb5793>
- 12 Bedford, T. *et al.* Integrating influenza antigenic dynamics with molecular evolution. *Elife* **3**, e01914 (2014). <https://doi.org/10.7554/eLife.01914>
- 13 Pei, S., Kandula, S., Yang, W. & Shaman, J. Forecasting the spatial transmission of influenza in the United States. *Proc Natl Acad Sci U S A* **115**, 2752-2757 (2018). <https://doi.org/10.1073/pnas.1708856115>
- 14 Goldstein, E., Cobey, S., Takahashi, S., Miller, J. C. & Lipsitch, M. Predicting the epidemic sizes of influenza A/H1N1, A/H3N2, and B: a statistical method. *PLoS Med* **8**, e1001051 (2011). <https://doi.org/10.1371/journal.pmed.1001051>
- 15 Goldstein, E., Viboud, C., Charu, V. & Lipsitch, M. Improving the estimation of influenza-related mortality over a seasonal baseline. *Epidemiology* **23**, 829-838 (2012). <https://doi.org/10.1097/EDE.0b013e31826c2dda>
- 16 Pei, S., Teng, X., Lewis, P. & Shaman, J. Optimizing respiratory virus surveillance networks using uncertainty propagation. *Nat Commun* **12**, 222 (2021). <https://doi.org/10.1038/s41467-020-20399-3>

- 17 Cori, A., Ferguson, N. M., Fraser, C. & Cauchemez, S. A new framework and software to estimate time-varying reproduction numbers during epidemics. *Am J Epidemiol* **178**, 1505-1512 (2013). <https://doi.org/10.1093/aje/kwt133>
- 18 Hale, T. *et al.* A global panel database of pandemic policies (Oxford COVID-19 Government Response Tracker). *Nat Hum Behav* **5**, 529-538 (2021). <https://doi.org/10.1038/s41562-021-01079-8>

REVIEWERS' COMMENTS

Reviewer #1 (Remarks to the Author):

I thank the authors for addressing my remarks and revising the manuscript accordingly. I understand that Meta's Colocation Maps for the study period are not publicly available, as I erroneously wrote in my original assessment. I appreciate the authors' effort to access the data although unsuccessful.

I think the manuscript has improved and can be accepted for publication in Nature Communications.

Reviewer #1 (Remarks on code availability):

I haven't reviewed the code, as I am not an expert user of R.

From what I see in the repository, a detailed explanation of how to use the code is missing but the README says "documentation forthcoming". I imagine the authors will provide more details upon publication.

Reviewer #2 (Remarks to the Author):

Thank you for your detailed response. The flow charts for pathogen incidence reconstruction and statistical analysis look great.

Reviewer #2 (Remarks on code availability):

The link can be opened. I only browsed the files, but did not test the code (and some data cannot be made public due to the data access agreement). Overall, the entire GitHub repository structure and Readme document are relatively clear and complete.